# Integrase-RNA interactions underscore the critical role of integrase in HIV-1 virion morphogenesis

Jennifer L Elliott[1], Jenna E Eschbach[1], Pratibha C Koneru[2], Wen Li[3,4], Maritza Puray-Chavez[1], Dana Townsend[1], Dana Q Lawson[1], Alan N Engelman[3,4], Mamuka Kvaratskhelia[2], Sebla B Kutluay[1]*

[1]Department of Molecular Microbiology, Washington University School of Medicine, Saint Louis, United States; [2]Division of Infectious Diseases, University of Colorado School of Medicine, Aurora, United States; [3]Department of Cancer Immunology and Virology, Dana-Farber Cancer Institute, Boston, United States; [4]Department of Medicine, Harvard Medical School, Boston, United States

**Abstract** A large number of human immunodeficiency virus 1 (HIV-1) integrase (IN) alterations, referred to as class II substitutions, exhibit pleiotropic effects during virus replication. However, the underlying mechanism for the class II phenotype is not known. Here we demonstrate that all tested class II IN substitutions compromised IN-RNA binding in virions by one of the three distinct mechanisms: (i) markedly reducing IN levels thus precluding the formation of IN complexes with viral RNA; (ii) adversely affecting functional IN multimerization and consequently impairing IN binding to viral RNA; and (iii) directly compromising IN-RNA interactions without substantially affecting IN levels or functional IN multimerization. Inhibition of IN-RNA interactions resulted in the mislocalization of viral ribonucleoprotein complexes outside the capsid lattice, which led to premature degradation of the viral genome and IN in target cells. Collectively, our studies uncover causal mechanisms for the class II phenotype and highlight an essential role of IN-RNA interactions for accurate virion maturation.

*For correspondence:
kutluay@wustl.edu

Competing interests: The authors declare that no competing interests exist.

## Introduction

Infectious HIV-1 virions are formed in a multistep process coordinated by interactions between the HIV-1 Gag and Gag-Pol polyproteins, and the viral RNA (vRNA) genome. At the plasma membrane of an infected cell, Gag and Gag-Pol molecules assemble around a vRNA dimer and bud from the cell as a spherical immature virion, in which the Gag proteins are radially arranged (*Sundquist and Kräusslich, 2012*; *Pornillos and Ganser-Pornillos, 2019*; *Bieniasz and Telesnitsky, 2018*). As the immature virion buds, the viral protease enzyme is activated and cleaves Gag and Gag-Pol into their constituent domains, triggering virion maturation (*Sundquist and Kräusslich, 2012*; *Pornillos and Ganser-Pornillos, 2019*). During maturation, the cleaved nucleocapsid (NC) domain of Gag condenses with the RNA genome and *pol*-encoded viral enzymes (reverse transcriptase [RT] and integrase [IN]) inside the conical capsid lattice, composed of the cleaved capsid (CA) protein, which together forms the core (*Sundquist and Kräusslich, 2012*; *Pornillos and Ganser-Pornillos, 2019*; *Bieniasz and Telesnitsky, 2018*).

After infection of a target cell, RT in the confines of the reverse transcription complex (RTC) synthesizes linear double-stranded DNA from vRNA (*Engelman, 2010*). The vDNA is subsequently imported into the nucleus, where the IN enzyme catalyzes its insertion into the host cell chromosome (*Engelman, 2019*; *Lesbats et al., 2016*). Integration is mediated by the intasome nucleoprotein complex that consists of a multimer of IN engaging both ends of linear vDNA (*Engelman and*

*Cherepanov, 2017*). While the number of IN protomers required for intasome function varies across Retroviridae, single-particle cryogenic electron microscopic (cryo-EM) structures of HIV-1 and Maedi-visna virus indicate that lentivirus integration proceeds via respective higher-order dodecamer and hexadecamer IN arrangements (*Ballandras-Colas et al., 2017*; *Passos et al., 2017*), though a lower-order intasome composed of an HIV-1 IN tetramer was also resolvable by cryo-EM (*Passos et al., 2017*).

A number of IN substitutions which specifically arrest HIV-1 replication at the integration step have been described (*Engelman, 1999*). These substitutions are grouped into class I to delineate them from a variety of other IN substitutions, which exhibit pleiotropic effects, and are collectively referred to as class II substitutions (*Engelman, 1999*; *Engelman et al., 1995*; *Engelman, 2011*). Class II IN substitutions or deletions of entire IN impair proper particle assembly (*Engelman et al., 1995*; *Ansari-Lari et al., 1995*; *Bukovsky and Göttlinger, 1996*; *Jenkins et al., 1996*; *Kalpana et al., 1999*; *Leavitt et al., 1996*; *Liao and Wang, 2004*; *Lu et al., 2005a*; *Lu et al., 2004*; *Nakamura et al., 1997*; *Quillent et al., 1996*; *Shin et al., 1994*; *Taddeo et al., 1994*; *Wu et al., 1999*), morphogenesis (*Engelman et al., 1995*; *Jenkins et al., 1996*; *Nakamura et al., 1997*; *Quillent et al., 1996*; *Shin et al., 1994*; *Fontana et al., 2015*; *Jurado et al., 2013*; *Kessl et al., 2016*), and reverse transcription in target cells (*Engelman, 1999*; *Engelman et al., 1995*; *Leavitt et al., 1996*; *Lu et al., 2005a*; *Lu et al., 2004*; *Nakamura et al., 1997*; *Shin et al., 1994*; *Wu et al., 1999*; *Fontana et al., 2015*; *Jurado et al., 2013*; *Kessl et al., 2016*; *Ao et al., 2005*; *Busschots et al., 2007*; *Engelman et al., 1997*; *Limón et al., 2002*; *Lloyd et al., 2007*; *Lu et al., 2005b*; *Masuda et al., 1995*; *Rahman et al., 2007*; *Rivière et al., 2010*; *Tsurutani et al., 2000*; *Wiskerchen and Muesing, 1995*; *Zhu et al., 2004*; *De Houwer et al., 2014*; *Johnson et al., 2013*; *Mohammed et al., 2011*; *Shehu-Xhilaga et al., 2002*), in some cases without impacting IN catalytic function in vitro (*Jenkins et al., 1996*; *Kalpana et al., 1999*; *Lu et al., 2005a*; *Lu et al., 2004*; *Busschots et al., 2007*; *Engelman et al., 1997*; *Lu et al., 2005b*; *Rahman et al., 2007*; *Engelman and Craigie, 1992*; *Lutzke and Plasterk, 1998*; *Lutzke et al., 1994*). A hallmark morphological defect of these viruses is the formation of aberrant viral particles with viral ribonucleoprotein (vRNP) complexes mislocalized outside the conical CA lattice (*Engelman et al., 1995*; *Jenkins et al., 1996*; *Nakamura et al., 1997*; *Quillent et al., 1996*; *Shin et al., 1994*; *Fontana et al., 2015*; *Jurado et al., 2013*; *Kessl et al., 2016*). Strikingly similar morphological defects are observed in virions produced from cells treated with allosteric integrase inhibitors (ALLINIs, also known as LEDGINs, NCINIs, INLAIs, or MINIs; *Fontana et al., 2015*; *Jurado et al., 2013*; *Balakrishnan et al., 2013*; *Sharma et al., 2014*; *Le Rouzic et al., 2013*; *Desimmie et al., 2013*; *Slaughter et al., 2014*; *Amadori et al., 2017*; *Gupta et al., 2014*; *Bonnard et al., 2018*). ALLINIs induce aberrant IN multimerization in virions by engaging the V-shaped pocket at the IN dimer interface, which also provides a principal binding site for the host integration targeting cofactor lens epithelium-derived growth factor (LEDGF)/p75 (*Le Rouzic et al., 2013*; *Gupta et al., 2014*; *Deng et al., 2016*; *Feng et al., 2013*; *Gupta et al., 2016*; *Koneru et al., 2019*; *Kessl et al., 2012*). The recent discovery that HIV-1 IN binds to the vRNA genome in virions and that inhibiting IN-RNA interactions leads to the formation of eccentric particles provided initial clues about the role of IN during virion morphogenesis (*Kessl et al., 2016*).

HIV-1 IN consists of three independently folded protein domains: the N-terminal domain (NTD), catalytic core domain (CCD), and C-terminal domain (CTD) (*Engelman and Cherepanov, 2017*; *Engelman and Cherepanov, 2014*), and vRNA binding is mediated by a constellation of basic residues within the CTD (*Kessl et al., 2016*). However, class II IN substitutions are located throughout the entire length of the IN protein (*Engelman, 1999*; *Engelman, 2011*), which raises the question as to how these substitutions impair virus maturation. The structural basis for IN binding to RNA is not yet known; however, in vitro evidence indicates that IN binds RNA as lower-order multimers, and conversely RNA binding may prevent the formation of higher-order IN multimers (*Kessl et al., 2016*). Notably, aberrant IN multimerization underlies the inhibition of IN-RNA interactions by ALLINIs (*Kessl et al., 2016*) and subsequent defects in virion maturation (*Fontana et al., 2015*; *Jurado et al., 2013*; *Kessl et al., 2016*; *Balakrishnan et al., 2013*; *Sharma et al., 2014*; *Desimmie et al., 2013*; *Slaughter et al., 2014*; *Amadori et al., 2017*; *Gupta et al., 2014*; *Bonnard et al., 2018*). Therefore, it seems plausible that class II IN substitutions may exert their effect on virus replication by adversely affecting functional IN multimerization. However, a systematic evaluation of the effects of IN substitutions on IN multimerization, IN-RNA binding, and virion

morphology is lacking. As such, it remains an open question as to how functional IN multimerization and/or IN-RNA interactions influence correct virion morphogenesis.

Eccentric virions generated via class II IN substitutions or ALLINI treatment are defective for reverse transcription in target cells (*Engelman, 1999*; *Engelman et al., 1995*; *Leavitt et al., 1996*; *Lu et al., 2005a*; *Lu et al., 2004*; *Nakamura et al., 1997*; *Shin et al., 1994*; *Wu et al., 1999*; *Fontana et al., 2015*; *Jurado et al., 2013*; *Kessl et al., 2016*; *Ao et al., 2005*; *Busschots et al., 2007*; *Engelman et al., 1997*; *Limón et al., 2002*; *Lloyd et al., 2007*; *Lu et al., 2005b*; *Masuda et al., 1995*; *Rahman et al., 2007*; *Rivière et al., 2010*; *Tsurutani et al., 2000*; *Wiskerchen and Muesing, 1995*; *Zhu et al., 2004*; *De Houwer et al., 2014*; *Johnson et al., 2013*; *Mohammed et al., 2011*; *Shehu-Xhilaga et al., 2002*; *Balakrishnan et al., 2013*; *Sharma et al., 2014*; *Desimmie et al., 2013*; *Gupta et al., 2014*; *Gupta et al., 2016*; *Tekeste et al., 2015*) despite containing equivalent levels of RT and vRNA genome as wild type (WT) particles (*Fontana et al., 2015*; *van Bel et al., 2014*). In addition, neither the condensation of the viral genome by NC (*Fontana et al., 2015*; *van Bel et al., 2014*) nor its priming (*van Bel et al., 2014*) appears to be affected. We and others have recently shown that premature loss of the viral genome and IN, as well as spatial separation of RT from vRNPs, may underlie the reverse transcription defect observed in eccentric viruses generated in the presence of ALLINIs or the class II IN R269A/K273A substitutions (*Koneru et al., 2019*; *Madison et al., 2017*). These findings support a model in which the capsid lattice or IN binding to vRNA itself is necessary to protect viral components from the host environment upon entering a target cell. Whether the premature loss of the viral genome and IN is a universal outcome of other class II IN substitutions is unknown.

In this study, we aimed to determine the molecular basis of how class II IN substitutions exert their effects on HIV-1 replication. In particular, by detailed characterization of how class II substitutions impact IN multimerization, IN-RNA interactions, and virion morphology, we aimed to dissect whether loss of IN binding to vRNA or aberrant IN multimerization underlies the pleiotropic defects observed in viruses bearing class II IN mutations. Remarkably, we found that class II substitutions either prevented IN binding to the vRNA genome or precluded the formation of IN-vRNA complexes through reducing or eliminating IN from virions. We show that IN tetramers have a strikingly higher affinity toward vRNA than IN monomers or dimers, and many class II IN substitutions inhibited IN binding to RNA indirectly through modulating functional IN tetramerization. By contrast, R262A/R263A and R269A/K273A substitutions within the CTD and the K34A change within the NTD did not perturb IN tetramer formation, and thus likely directly interfered with IN binding to RNA. Irrespective of how IN-RNA binding was inhibited, all class II IN mutant viruses formed eccentric particles with vRNPs mislocalized outside the CA lattice. Subsequently, this led to premature loss of the vRNA genome as well as IN, and spatial separation of RT and CA from the vRNPs in target cells. Taken together, our findings uncover causal mechanisms for the class II phenotype and highlight the essential role of IN-RNA interactions for the formation of correctly matured virions and vRNP stability in HIV-1-infected cells.

## Results

### Characterization of the replication defects of class II IN mutant viruses

Substitutions in IN that exhibited a class II phenotype i.e. assembly, maturation or reverse transcription defects (*Engelman, 1999*; *Engelman et al., 1995*; *Engelman, 2011*; *Ansari-Lari et al., 1995*; *Bukovsky and Göttlinger, 1996*; *Jenkins et al., 1996*; *Kalpana et al., 1999*; *Leavitt et al., 1996*; *Liao and Wang, 2004*; *Lu et al., 2005a*; *Lu et al., 2004*; *Nakamura et al., 1997*; *Quillent et al., 1996*; *Shin et al., 1994*; *Taddeo et al., 1994*; *Wu et al., 1999*; *Fontana et al., 2015*; *Jurado et al., 2013*; *Kessl et al., 2016*; *Ao et al., 2005*; *Busschots et al., 2007*; *Engelman et al., 1997*; *Limón et al., 2002*; *Lloyd et al., 2007*; *Lu et al., 2005b*; *Masuda et al., 1995*; *Rahman et al., 2007*; *Rivière et al., 2010*; *Tsurutani et al., 2000*; *Wiskerchen and Muesing, 1995*; *Zhu et al., 2004*; *De Houwer et al., 2014*; *Johnson et al., 2013*; *Mohammed et al., 2011*; *Shehu-Xhilaga et al., 2002*; *Englund et al., 1995*; *Petit et al., 1999*) or affected IN multimerization (*Lutzke and Plasterk, 1998*; *Eijkelenboom et al., 1999*; *Hare et al., 2009*; *Kessl et al., 2009*; *Li et al., 2012*) were selected from past literature. Although a structure of IN bound to RNA is not currently available, the location of these substitutions depicted on the model of a tetrameric IN complex (based on

the cryo-EM structure of the HIV-1 intasome complex consisting of IN and DNA [*Passos et al., 2017*]) suggest that the targeted amino acids are positioned at or near monomer-monomer or dimer-dimer interfaces (*Figure 1A–B*). While not apparent in the tetrameric intasome complex, the CTD mediates IN tetramer-tetramer interactions in the higher-order dodecamer IN structure (*Passos et al., 2017*) and has also been shown to mediate IN multimerization in vitro (*Jenkins et al., 1996*).

IN mutations were introduced into the replication-competent pNL4-3 molecular clone and HEK293T cells were transfected with the resulting plasmids. Cell lysates and cell-free virions were subsequently analyzed for Gag/Gag-Pol expression, processing, particle release, and infectivity. While substitutions in IN had no measurable effect on Gag (Pr55) expression, modest effects on Gag processing in cells was visible for several missense mutant viruses including H12N, N18I, K34A, Y99A, K103E, W108R, F185K, Q214L/Q216L, L242A, V260E, as well as the ΔIN mutant (*Figure 2A*).

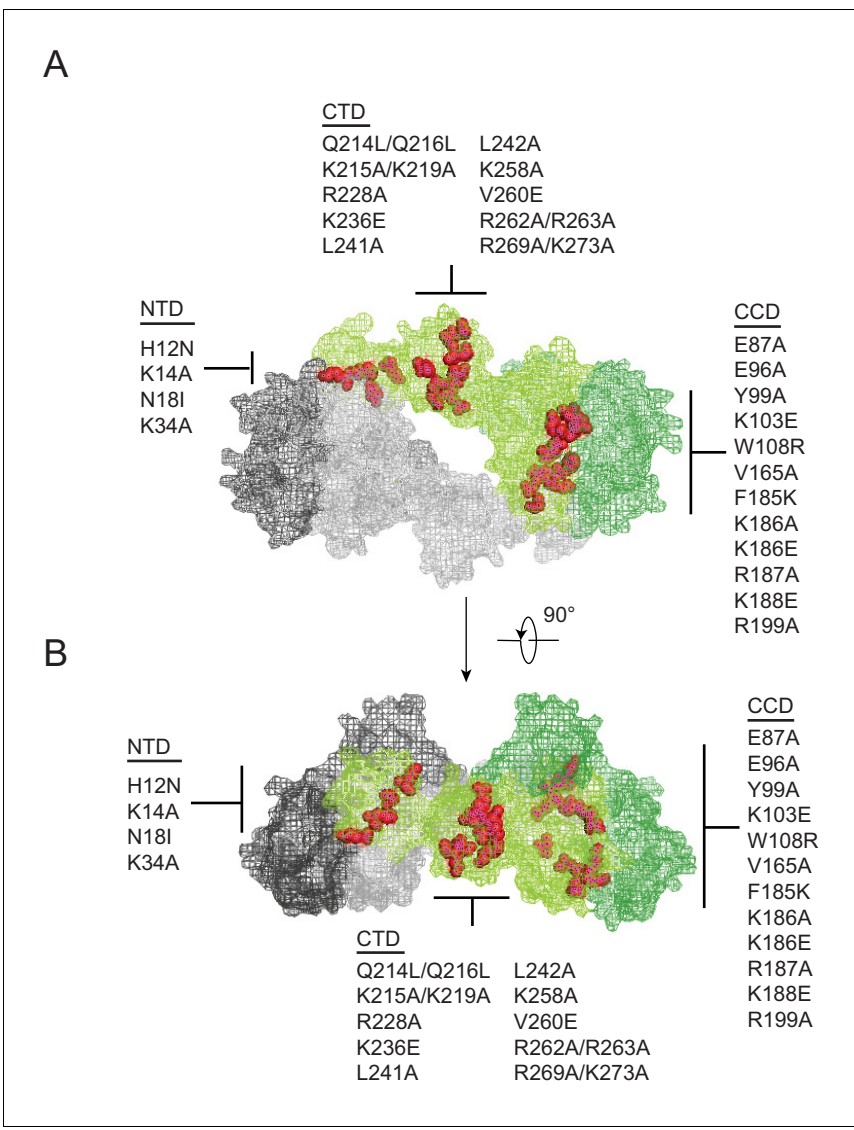

**Figure 1.** Class II IN substitutions locate throughout IN and cluster at interfaces that mediate IN multimerization. (**A**) Location of class II IN substitutions used in this study displayed in red on a single IN monomer within the context of the HIV-1 IN tetramer intasome structure consisting of a dimer of dimers (PDB 5U1C). The two dimers are displayed in either gray or green, with individual monomers within each displayed in different shades. The DNA is omitted for clarity. (**B**) View of the structure displayed in A rotated 90˚.

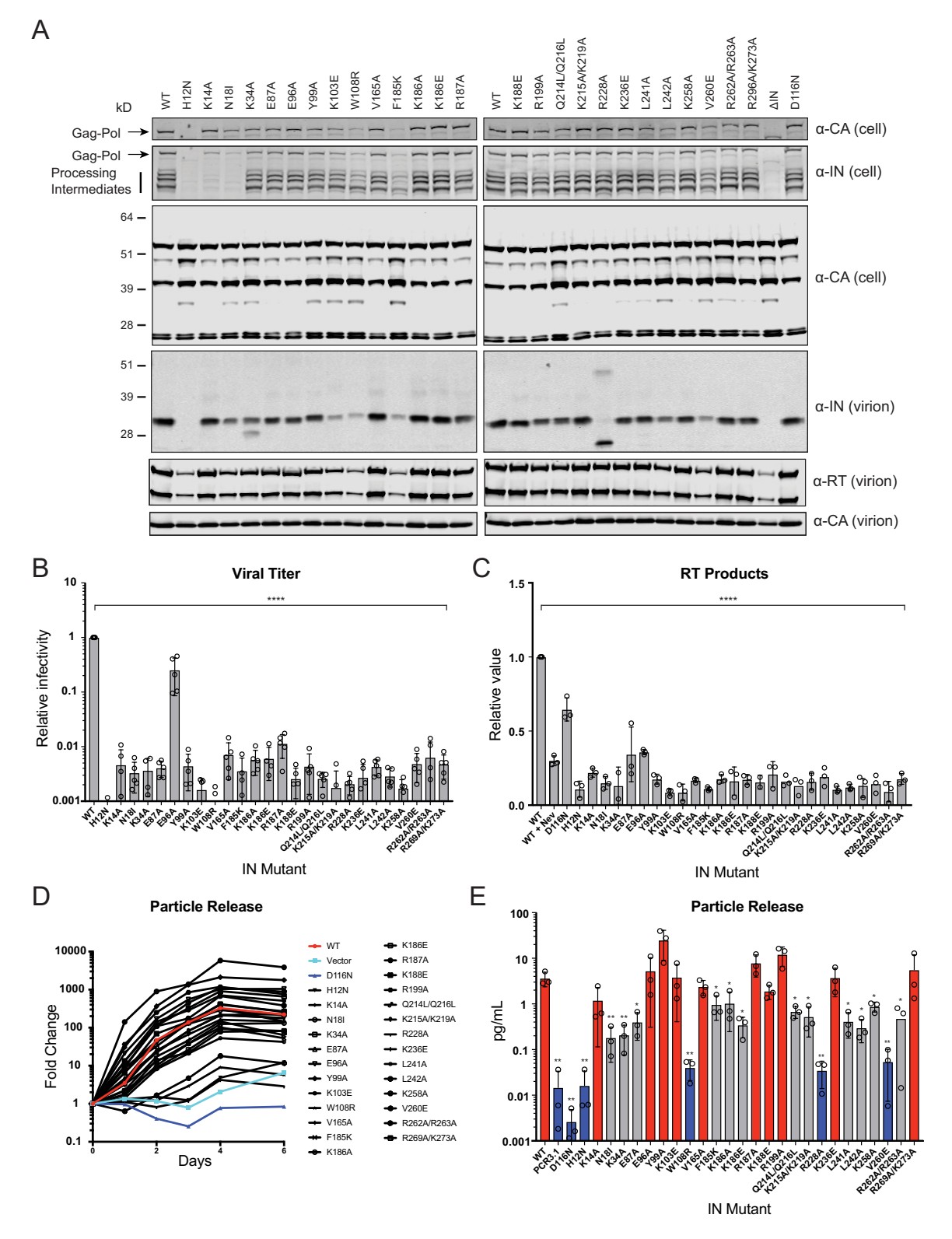

**Figure 2.** Characterization of the replication defects of class II IN mutant viruses. (A) Immunoblot analysis of Gag and Gag-Pol products in cell lysates and virions. HEK293T cells were transfected with proviral HIV-1_NL4-3 expression plasmids carrying *pol* mutations encoding for the indicated IN substitutions. Cell lysates and purified virions were harvested 2 d post-transfection and analyzed by immunoblotting for CA, IN, and, in the case of virions, RT. A representative image of one of four independent experiments is shown. (B) Infectious titers of WT or IN mutant HIV-1_NL4-3 viruses in cell

*Figure 2 continued on next page*

*Figure 2 continued*

culture supernatants were determined on TZM-bl indicator cells. Titer values are expressed relative to WT (set to 1). Columns show the average of five independent experiments (open circles) and error bars represent standard deviation (****p<0.0001, by one-way ANOVA with Dunnett's multiple comparison test). (C) The relative quantity of reverse-transcribed HIV-1 DNA in MT-4 target cells infected with HIV-1$_{NL4-3}$ at 6 hpi. Quantities of vDNA are expressed relative to WT (set to 1). Columns show the average of three independent experiments (open circles) and error bars represent standard deviation (****p<0.0001, by one-way ANOVA with Dunnett's multiple comparison test). (D) A representative growth curve of HIV-1$_{NL4-3}$ IN$_{D116N}$ viruses trans-complemented with class II mutant IN proteins in cell culture. Y-axis indicates fold increase in virion yield over day 0 as measured by RT activity in culture supernatants. HIV-1$_{NL4-3}$ IN$_{D116N}$ viruses that were trans-complemented with WT IN, class II mutant INs, IN$_{D116N}$, or an empty vector are denoted as red, black, dark blue, and light blue lines respectively. A representative plot from one of three independent experiments. (E) Fold increase in virions in culture supernatants at 4 dpi, as measured by RT activity in culture supernatants. Trans-complementation of the HIV-1$_{NL4-3}$ IN$_{D116N}$ virus with mutant IN molecules restored particle release to levels comparable to WT IN (red), partially restored particle release (gray) or could not restore particle release (blue). Columns show the average of three independent experiments (open circles) and error bars represent standard deviation (*p<0.05 and **p<0.01, by paired t-test between individual mutants and WT).

The online version of this article includes the following figure supplement(s) for figure 2:

**Figure supplement 1.** Characterization of the replication defects of class II IN mutant viruses.

Nevertheless, particle release was largely similar between WT and IN mutant viruses, as evident by the similar levels of CA protein present in cell culture supernatants (*Figure 2A*, lower panels).

Three distinct phenotypes became apparent by assessing the amount of virion-associated IN and RT enzymes (*Figure 2A*, *Figure 2—figure supplement 1A*). First, virion-associated IN was at least 5-fold less than WT with several mutants, including H12N, N18I, K103E, W108R, F185K, L242A, and V260E (*Figure 2A* and *Supplementary file 1*). Notably, these substitutions also reduced levels of Gag-Pol processing intermediates in producer cells (*Figure 2A*) and RT in virions (*Figure 2A*, *Figure 2—figure supplement 1A*), suggesting that they likely destabilized the Gag-Pol precursor. Near-complete lack of processing intermediates with the K14A and N18I substitutions, despite the presence of fully processed RT and IN in virions (detected using a separate polyclonal antibody), is likely due to the inaccessibility of epitopes recognized by the monoclonal anti-IN antibody in the processing intermediates. Second, the R228A substitution abolished full-length IN in virions without impacting cell- or virion-associated Gag-Pol levels or processing intermediates; however, a faster migrating species generated by aberrant IN processing and/or IN degradation was visible. A similar but more modest defect was observed for the K34A mutant, which was incorporated into virions at a modestly reduced level alongside a smaller protein species. Third, the remainder of the IN substitutions did not appear to affect IN or Gag-Pol levels in cells or virions.

With the exception of E96A, nearly all of the IN substitutions reduced virus titers at least 100-fold compared to the WT (*Figure 2B*), which corresponded with reduced levels of reverse-transcription in infected cells (*Figure 2C*). In line with previous reports (*Lu et al., 2005a*; *Lu et al., 2004*; *Lu et al., 2005b*), class II mutant IN molecules had variable levels of catalytic activity as assessed by the ability of Vpr-IN proteins to transcomplement a catalytically inactive IN (D116N, *Engelman et al., 1995*; *Engelman and Craigie, 1992*) in infected cells (*Liu et al., 1997*; *Fletcher et al., 1997*). All Vpr-IN fusion proteins, except for the H12N mutant which likely decreased the stability of the Vpr-IN fusion protein, were expressed at similar levels in cells (*Figure 2—figure supplement 1B*). We found that K14A, E96A, Y99A, K103A, V165A, R187A, K188E R199A, K236E, and R269A/R273A IN mutants trans-complemented a catalytically inactive IN at levels similar to the WT, whereas W108R, R228A, and V260E mutants were unable to do so (*Figure 2D–E*). The inability of W108R, R228A, and V260E mutants to transcomplement implies that they are impaired for integration, a result in line with previous observations (*Lutzke and Plasterk, 1998*; *Li et al., 2012*). The remainder of the IN mutants restored integration, albeit at significantly lower than WT levels (*Figure 2D–E*). These results suggest that the majority of the class II mutant INs retain structural integrity and at least partial catalytic activity in the presence of a complementing IN protein. Cumulatively, these data show that some class II substitutions in IN can affect the stability and/or processing of virion-associated proteins, but they all universally lead to the formation of non-infectious virions that are blocked at reverse transcription in target cells, a hallmark of class II IN substitutions (*Engelman, 1999*; *Engelman, 2011*).

## Class II IN mutants abolish IN binding to RNA

Using complementary in vitro and CLIP-based approaches, we have previously shown that viral genomic RNAs (vRNA) constitute the primary RNA species bound by IN in virions and that IN interacts with the viral genome through multiple basic residues (i.e. K264, K266, R269, K273) in its CTD (*Kessl et al., 2016*). In addition, IN-RNA interactions could also depend on proper IN multimerization, as ALLINI-induced aberrant IN multimerization potently inhibited the ability of IN to bind RNA (*Kessl et al., 2016*). Based on this, in the next set of experiments, we aimed to determine whether class II IN mutants bind vRNA, and if not, whether improper IN multimerization may underlie this defect.

IN-vRNA complexes were immunoprecipitated from UV-crosslinked virions and the levels of coimmunoprecipitating vRNA were assessed. Note that substitutions that significantly reduced the amount of IN in virions (*Figure 2A*, *Supplementary file 1*) were excluded from these experiments. All class II IN mutant viruses contained similar levels of vRNA, ruling out any inadvertent effects of the alterations on RNA packaging (*Figure 3A*). While the catalytically inactive IN D116N bound vRNA at a level that was comparable to the WT, nearly all of the class II IN mutant proteins failed to bind vRNA (*Figure 3B*). The E96A substitution, which had a fairly modest effect on virus titers as compared to other IN mutants (*Figure 2B*), decreased but did not abolish the ability of IN to bind RNA (*Figure 3B*). Thus, lack of RNA binding ability is a surprisingly common property of a disperse set of class II IN mutants, despite the fact that many of the altered amino acid residues are distally located from the CTD.

## IN multimerization plays a key role in RNA binding

As it seemed unlikely that all of the class II IN substitutions directly inhibited IN binding to RNA, we reasoned that they might indirectly abolish binding by perturbing proper IN multimerization. To test whether class II IN substitutions altered IN multimerization in a relevant setting, purified HIV-1$_{NL4-3}$ virions were treated with ethylene glycol bis(succinimidyl succinate) (EGS) to covalently crosslink IN in situ and virus lysates were analyzed by immunoblotting. IN species that migrated at molecular weights consistent with those of monomers, dimers, trimers, and tetramers were readily distinguished in WT virions (*Figure 4—figure supplement 1*). In the majority of the class II mutant particles, IN appeared to exist as monomers as well as higher molecular weight species, representing IN multimers or IN aggregates, with little dimers and no readily detectable tetramers (*Figure 4A*, *Figure 4—figure supplement 1*). In contrast, K34A, E96A, R262A/R263A, and R269A/K273A IN

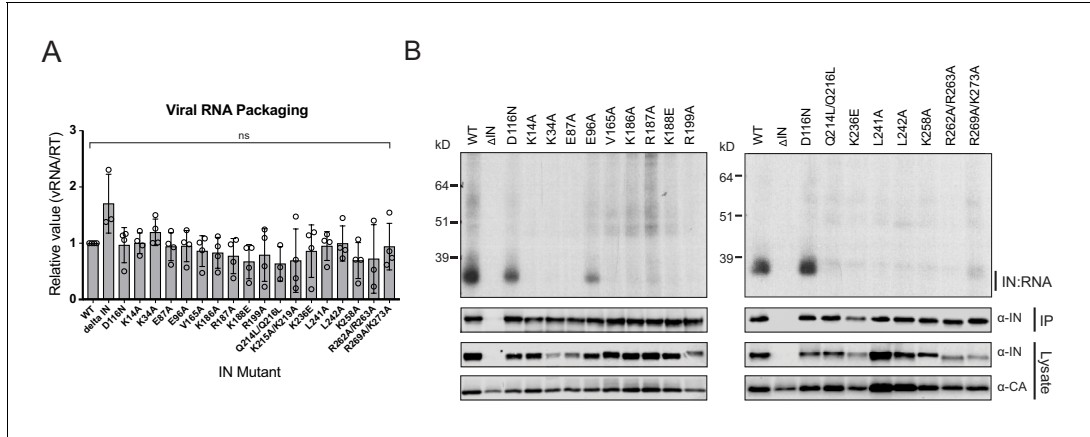

**Figure 3.** Class II IN substitutions prevent IN binding to the vRNA genome in virions. (**A**) Analysis of the levels of packaged viral genomic RNA in WT and IN mutant HIV-1$_{NL4-3}$ virions. vRNA extracted from purified virions was measured by Q-PCR. Data were normalized to account for differences in particle yield using an RT activity assay. Normalized quantities of vRNA are expressed relative to WT (set to 1). Columns show the average of three-four independent experiments (open circles) and error bars represent standard deviation (ns, not significant, by one-way ANOVA). (**B**) A representative autoradiogram of IN-RNA adducts immunoprecipitated from WT or IN mutant HIV-1$_{NL4-3}$ virions. The amount of immunoprecipitated material was normalized such that equivalent levels of WT and mutant IN proteins were loaded on the gel, as also evident in the immunoblots shown below. Levels of IN and CA in input virion lysates is shown in the lower immunoblots. Results are a representative of three independent replicates.

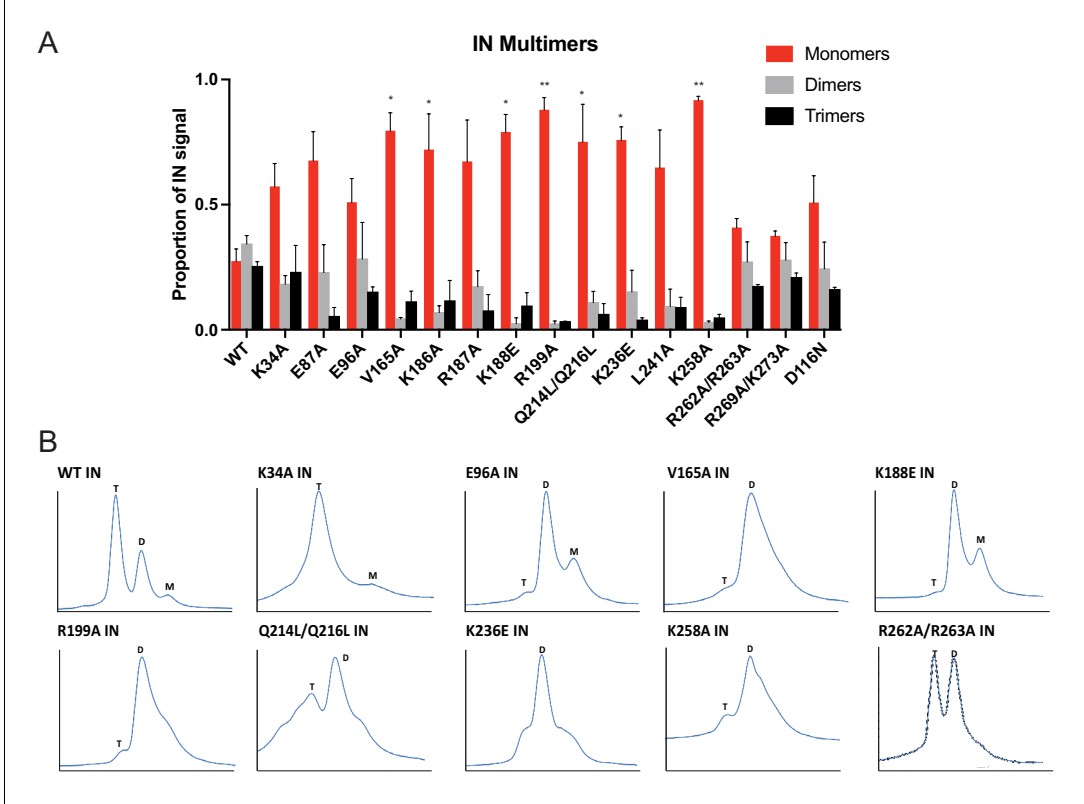

**Figure 4.** Multimerization properties of class II IN mutants in virions and in vitro. (**A**) Purified HIV-1 virions were crosslinked with 1 mM EGS and analyzed by immunoblotting as detailed in Materials and Methods. The IN signal at the molecular weights of 32 kDa (monomers), 64 kDa (dimers), and 96 kDa (trimers) was measured and divided by the total signal of the three multimeric species for each virus. Columns show the average of three independent experiments and error bars represent standard error of the mean (**p<0.05 and *p<0.01, by one-way ANOVA with Dunnett's multiple comparison test). (**B**) SEC profiles of 10 µM of WT and indicated IN mutants are shown. The X-axis indicates elution volume (mL) and Y-axis indicates the intensity of absorbance (mAU). Tetramers (T), dimers (D), and monomers (M) are indicated. Representative chromatograms from two independent analyses are shown.

The online version of this article includes the following figure supplement(s) for figure 4:

**Figure supplement 1.** Multimerization properties of class II IN mutants.

mutants formed dimers and tetramers at similar levels to the WT (*Figure 4A*, *Figure 4—figure supplement 1*). An undefined smear was present at higher molecular weights for all viruses, possibly as a result of the formation of large IN aggregates upon cross-linking (*Figure 4—figure supplement 1*).

To corroborate these findings, we analyzed the oligomeric states of recombinant WT, K34A, E96A, V165A, K188E, R199A, Q214L/Q216L, K236E, K258A, and R262A/R263A IN proteins by SEC (*Figure 4B*). Oligomeric states of additional class II IN mutants have previously been characterized in vitro (*Kessl et al., 2016*; *Lutzke and Plasterk, 1998*; *Hare et al., 2009*; *Kessl et al., 2009*; *McKee et al., 2008*; *Pandey et al., 2011*; *De Houwer et al., 2012*) and are summarized in *Supplementary file 2*. In line with the crosslinking studies in virions, WT, K34A, and R262A/R263A INs formed tetramers, while the levels of dimers varied for different mutants. For example, while IN R262A/R263A presented similar levels of tetramers and dimers, IN K34A was primarily tetrameric with a minor dimeric species, as evident by the broad right shoulder of the tetrameric SEC peak (*Figure 4B*). By contrast, the majority of other IN mutants almost exclusively formed dimers and monomers with little evidence for tetramer formation (*Figure 4B*). While Q214L/Q216L and K236E IN were predominantly dimeric, the broad base of their chromatograms revealed some evidence for tetramers and monomers as well (*Figure 4B*).

Next, we tested the mutant INs for their ability to bind and bridge cognate RNA oligonucleotides in vitro. We have previously shown that recombinant IN binds TAR RNA with a high affinity and

provides a nucleation point to bridge and condense RNA (*Kessl et al., 2016*). However, IN oligomeric states required for its ability to bind and bridge RNA molecules are not known. We have separated WT monomeric, dimeric, and tetrameric forms of WT IN by SEC and examined their binding to TAR RNA (*Kessl et al., 2016*). Remarkably, while WT IN tetramers bound to TAR RNA with high affinity (2.68 ± 0.16 nM), neither IN dimers nor monomers showed evidence of binding (*Figure 5A*). In line with this, we found that WT tetramers rather than dimers effectively bridged RNA oligonucleotides in vitro (*Figure 5B*).

We then analyzed a set of class II IN mutants for their ability to bind and bridge TAR RNA in vitro (*Figure 4B*). All class II IN mutants that predominantly formed dimers (*Figure 4B*) had reduced affinity for RNA compared to WT IN (*Figure 5C*, *Figure 5—figure supplement 1B–D*). Furthermore, these mutations had even more deleterious effects on the ability of IN to bridge the RNA molecules (*Figure 5C*). Although IN K34A and IN R262A/R263A could both form tetramers, IN K34A showed a reduced binding affinity for RNA while IN R262A/R263A did not bind RNA at all (*Figure 5C*, *Figure 5—figure supplement 1A*), suggesting that these residues may be directly involved in IN binding to RNA. Collectively, these results pointed to a key role of IN tetramerization for high-affinity binding to RNA and more critically for RNA bridging. Thus, these findings suggest that a defect in proper multimerization underlies the inability of the majority of class II IN mutants to form functional complexes with vRNA.

## Class II IN substitutions generate virions with eccentric morphology

We next sought to determine how preclusion or inhibition of IN-vRNA interactions correlated with particle morphology. Virion morphology of a subset of IN mutants that inhibited vRNA interactions by three different mechanisms; that is, those that decreased IN levels in virions (N18I and W108R), those that may have directly inhibited IN binding to RNA (K34A, R262A/R263A), and those that primarily altered IN multimerization (E87A, E96A, F185K, R187A, L241A, L242A), was assessed by transmission electron microscopy (TEM). As expected, the majority of WT particles contained an electron-dense condensate representing vRNPs inside the CA lattice, whereas an ΔRT-IN deletion mutant virus produced similar levels of immature particles and eccentric particles (*Figure 6A–B*). Remarkably, irrespective of how IN-RNA interactions were inhibited, 70–80% of nearly all class II IN mutant particles exhibited an eccentric morphology (*Figure 6A–B*). Of note, the E96A mutant

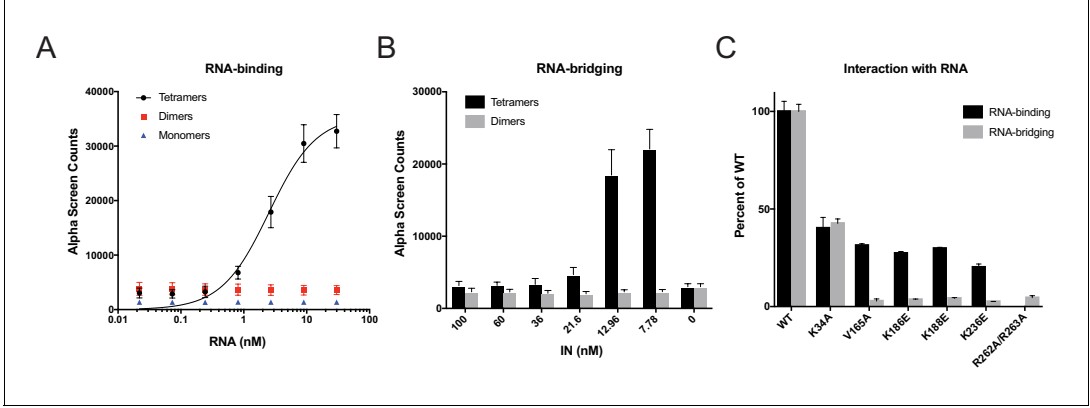

**Figure 5.** RNA-binding properties of class II IN mutants. (**A**) Analysis by AlphaScreen assay of 100 nM WT IN monomers, dimers, and tetramers binding to biotinylated TAR RNA after separation by SEC. Graphed data is the average of three independent experiments and error bars indicate standard deviation. (**B**) Summary of WT IN dimers and tetramers bridging TAR RNA at different protein concentrations as measured by AlphaScreen assay. Graphed data is the average of four independent experiments and error bars indicate standard deviation. (**C**) Summary of mutant INs binding to TAR RNA (black bars) and bridging TAR RNA (gray bars) compared to WT IN. Percent binding was calculated for each mutant IN by comparing the calculated Kd value to that of WT IN (normalized to 100%) within an experiment. Percent bridging was calculated by comparing the Alpha Counts at 320 nM for each protein to that of WT (normalized to 100%). Graphed data is the average of three independent experiments and error bars represent standard deviation.

The online version of this article includes the following figure supplement(s) for figure 5:

**Figure supplement 1.** RNA-binding properties of Class II IN mutants in vitro.

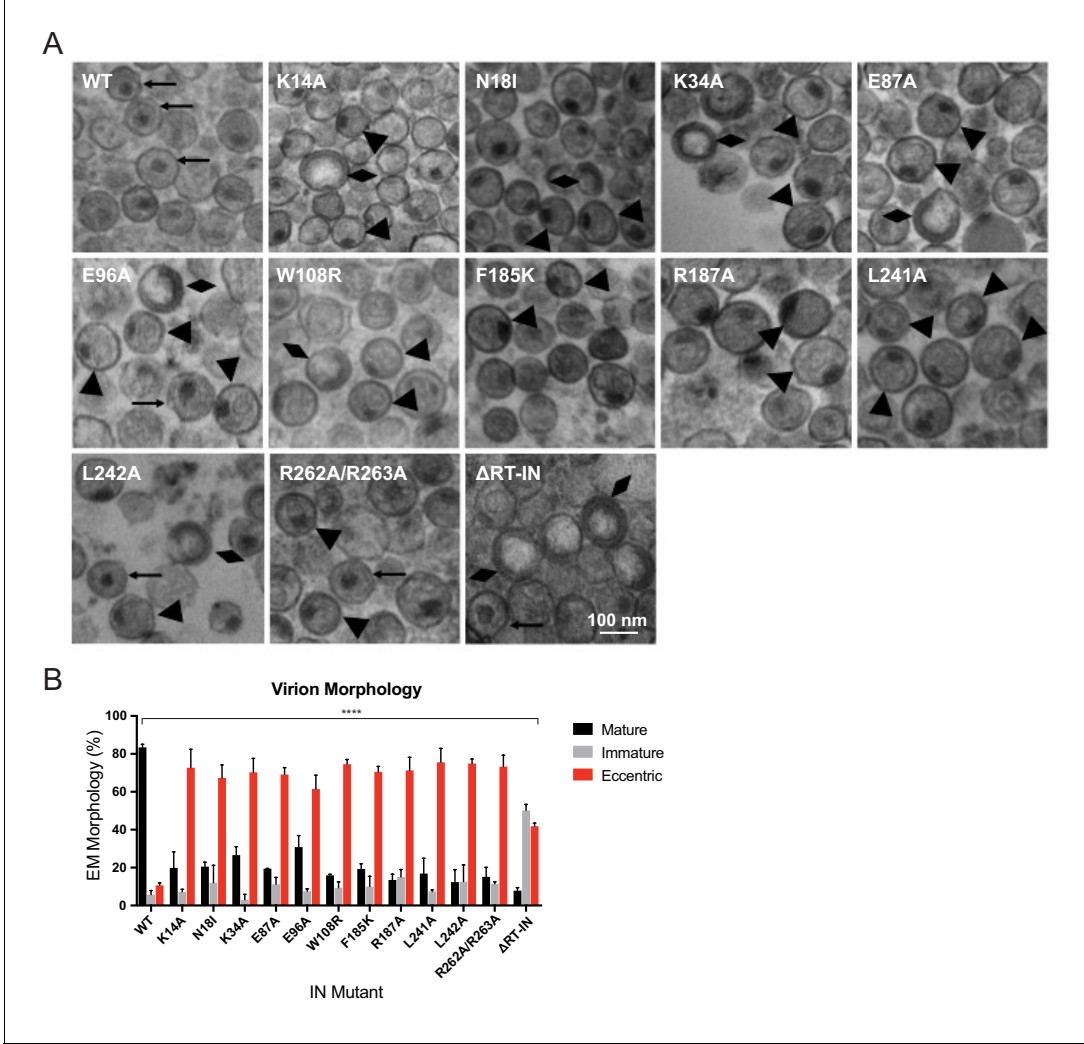

**Figure 6.** Analysis of class II IN mutant virion morphologies viruses by TEM. (**A**) Representative TEM images of WT, K14A, N18I, K34A, E87A, E96A, W108R, F185K, R187A, L241A, L242A, R262A/R263A, and ΔRT-IN HIV-1$_{NL4-3}$ virions. Magnification is 30,000× (scale bar, 100 nm). Black arrows indicate mature particles containing conical or round cores with associated electron density; triangles indicate eccentric particles with electron-densematerial situated between translucent cores and the viral membrane; diamonds indicate immature particles. (**B**) Quantification of virion morphologies. Columns show the average of two independent experiments (more than 100 particles counted per experiment) and error bars represent standard deviation (****p<0.0001, by repeated measures one-way ANOVA.).

tended to produce less eccentric and more mature particles than the other IN mutants. Because IN E96A retained partial binding to vRNA in virions (*Figure 3B*) and partial infectivity (*Figure 2B*), we conclude that this infection-deferred mutant harbors a partial class II phenotype.

Next, we tested whether inhibition of IN-RNA interactions through class II substitutions changes the localization of IN in virions. The premise for this is based on our previous finding that disruption of IN binding to vRNA through the IN R269A/K273A substitution leads to separation of a fraction of IN from dense vRNPs and CA containing complexes (*Madison et al., 2017*). Thus, we predicted that inhibition of IN-RNA interactions through the above class II substitutions could lead to a similar outcome. To this end, WT or class II IN mutant virions stripped of the viral lipid envelope by brief detergent treatment were separated on sucrose gradients, and resulting fractions were analyzed for CA, IN, and matrix (MA) content by immunoblotting (*Madison et al., 2017*; *Welker et al., 2000*). As before (*Madison et al., 2017*), WT IN migrated primarily in dense fractions, whereas the R269A/K273A mutant migrated bimodally (*Figure 7A,B*). In contrast to our hypothesis, the majority of IN mutants sedimented similarly to WT IN and settled in the denser gradient fractions (*Figure 7A,B*).

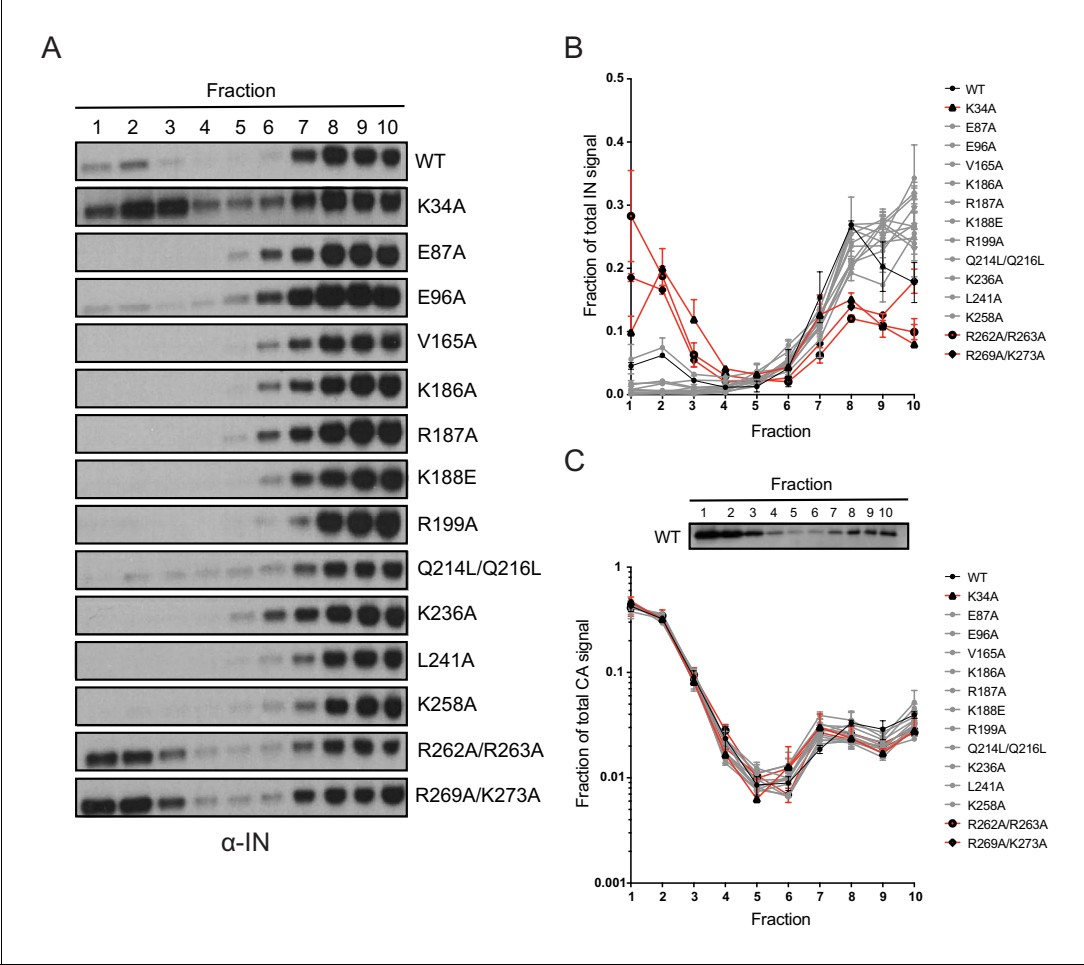

**Figure 7.** Biochemical analysis of class II IN mutant virus particles. (A) Immunoblot analysis of sedimentation profiles of IN in WT or IN mutant virions. Purified HIV-1$_{NLGP}$ virions were analyzed by equilibrium density centrifugation as detailed in Materials and Methods. Ten fractions collected from the top of the gradients were analyzed by immunoblotting using antibodies against IN. Representative images from one of four independent experiments are shown. (B) Quantitation of IN signal intensity in immunoblots as in (A) is shown. Profile of WT virions is denoted in black, IN mutants that led to bimodal IN distribution are shown in red and others are shown in gray. Graphed data is the average of two independent experiments and error bars indicate the range. (C) The representative immunoblot analysis of the sedimentation profile of CA in WT virions and quantitation of CA signal intensity in immunoblots are shown. Profile of WT virions is denoted in black, IN mutants that led to bimodal IN distribution are shown in red and others are shown in gray. Graphed data is the average of two independent experiments and error bars indicate the range.

The online version of this article includes the following figure supplement(s) for figure 7:

**Figure supplement 1.** Biochemical properties of Class II IN mutants in virions upon CA destabilization and in the absence of NC.

Exceptions were the K34A and R262A/R263A IN mutants, a fraction of which migrated to soluble fractions similar to the R269A/K273A mutant, suggesting their localization outside of the capsid lattice. None of the IN substitutions affected the migration pattern of CA (*Figure 7C*), which distributed was bimodally between the soluble and dense fractions, nor the distribution of MA (data not shown), which was found in mainly the soluble fractions. These results suggested that, with the exception of the K34A, R262A/R263A, and R269A/K273A, IN mutant proteins may remain associated with the CA lattice despite inhibition of IN-vRNA interactions. Alternatively, class II IN mutants may localize outside of the CA lattice but form aggregates resulting in a similar migration pattern in dense fractions.

To test these possibilities, we combined the CA destabilizing K203A substitution, which leads to premature disassembly of the CA lattice in vitro (*Forshey et al., 2002*), with class II IN substitutions K34A, E87A, and R262A/R263A. Biochemical fractionation of these viruses following membrane stripping as above yielded a similar IN distribution primarily in dense fractions (*Figure 7—figure*

*supplement 1A*). Note that the K203A CA mutation caused the loss of CA in the dense fractions compared to WT viruses, indicating that the CA lattice was indeed destabilized (*Figure 7—figure supplement 1A*). Thus, the migration of IN in dense fractions over the gradients does not appear to be dependent on its being enclosed in an intact CA lattice. We then asked if the mutant IN molecules may settle in the dense fractions due to some residual RNA binding not detected by the CLIP assay (*Figure 3B*). To test this, we introduced the above class II IN substitutions into a Gag-chimeric virus (Gag-bZIP) in which the NC domain is replaced by the leucine zipper domain from the yeast GCN4 protein and as a result is devoid of RNA (*Johnson et al., 2002*; *Accola et al., 2000*). Of note, this modification additionally destabilized the CA lattice, as indicated by the lack of a second population of CA in dense fractions (*Figure 7—figure supplement 1B*). Lack of RNA packaging and an unstable CA lattice with the Gag-bZIP virus also did not affect the migration pattern of IN over the sucrose gradients (*Figure 7—figure supplement 1B*), nor did presence of RNAse throughout the fractionation experiment (data not shown). The results from these experiments indicate that the migration of IN in dense sucrose gradients is independent of its being enclosed in an intact CA lattice and of its binding to vRNA.

## Premature loss of vRNA and IN from class II IN mutant viruses upon infection of target cells

We have previously shown that vRNA and IN are prematurely lost from cells infected with the R269A/K273A class II IN mutant (*Madison et al., 2017*). Given that eccentric vRNP localization is a common feature of class II IN mutant viruses (*Figure 6*), we next asked whether loss of vRNA in target cells is a common outcome for other class II IN mutant viruses. As the majority of mutant IN molecules appeared to remain associated with higher-order CA in virions (*Figure 7*), we also wanted to test whether they would be protected from premature degradation in infected cells.

The fates of viral core components in target cells were tracked using a previously described biochemical assay (*Kutluay et al., 2013*). For these experiments we utilized pgsA-745 cells (pgsA), which lack surface glycosaminoglycans, and likely as a result can be very efficiently infected by VSV-G-pseudotyped viruses in a synchronized fashion. PgsA cells were infected with WT or IN mutant viruses bearing substitutions that inhibited IN-vRNA interactions directly and may lead to mislocalization of IN in virions (i.e. K34A, R262A/R263A, R269A/K273A) or indirectly through aberrant IN multimerization and did not appear to grossly affect IN localization in virions (i.e. E87A, V165A; *Figure 8A*). Following infection, post-nuclear lysates were separated on linear sucrose gradients, and fractions collected from gradients were analyzed for viral proteins (CA, IN, RT) and vRNA by immunoblotting and Q-PCR-based assays, respectively.

As previously reported (*Madison et al., 2017*; *Kutluay et al., 2013*), in cells infected with WT viruses, IN, RT, vRNA, and a fraction of CA comigrated to sucrose fractions 6–8, representing active RTCs (*Figure 8B–E*). Note that a large fraction of CA migrated to the top two soluble sucrose fractions representing CA that had dissociated from the core as a result of uncoating or CA that was packaged into virions but not incorporated into the capsid lattice (*Briggs et al., 2004*; *Ganser-Pornillos et al., 2007*). Notably, in cells infected with class II IN mutant viruses, equivalent levels of CA (*Figure 8B*) and RT (*Figure 8D*) remained in the denser fractions, whereas IN (*Figure 8C*) and vRNA (*Figure 8E*) were substantially reduced. Loss of vRNA and IN from dense fractions, without any corresponding increase in the top fractions containing soluble proteins and RNA, suggest their premature degradation and/or mislocalization in infected cells.

Next, we employed a complementary microscopy-based assay (*Puray-Chavez et al., 2017*) in the context of full-length viruses to corroborate these findings. The advantages of this approach over biochemical fractionation experiments include the ability to track HIV-1 vRNA at the single-cell level with a high degree of specificity (*Figure 8—figure supplement 1A*), determine its subcellular localization, and to side-step possible post-processing artifacts associated with biochemical fractionation. Cells were synchronously infected with VSV-G pseudotyped HIV-1$_{NL4-3}$ in the presence of nevirapine to prevent vRNA loss due to reverse transcription, and vRNA levels associated with cells immediately following synchronization (0 hr) and 2 hr post-infection were evaluated (*Puray-Chavez et al., 2017*). In WT-infected cells, vRNA was visible immediately after infection (*Figure 8F*). Two hr post infection, cell-associated vRNA had fallen to 60–80% of starting levels (*Figure 8F*, *Figure 8—figure supplement 1C*), likely as the result of some viruses failing to enter or perhaps being degraded after entry. However, a significant proportion of vRNA was still readily detectable. By contrast, in cells infected

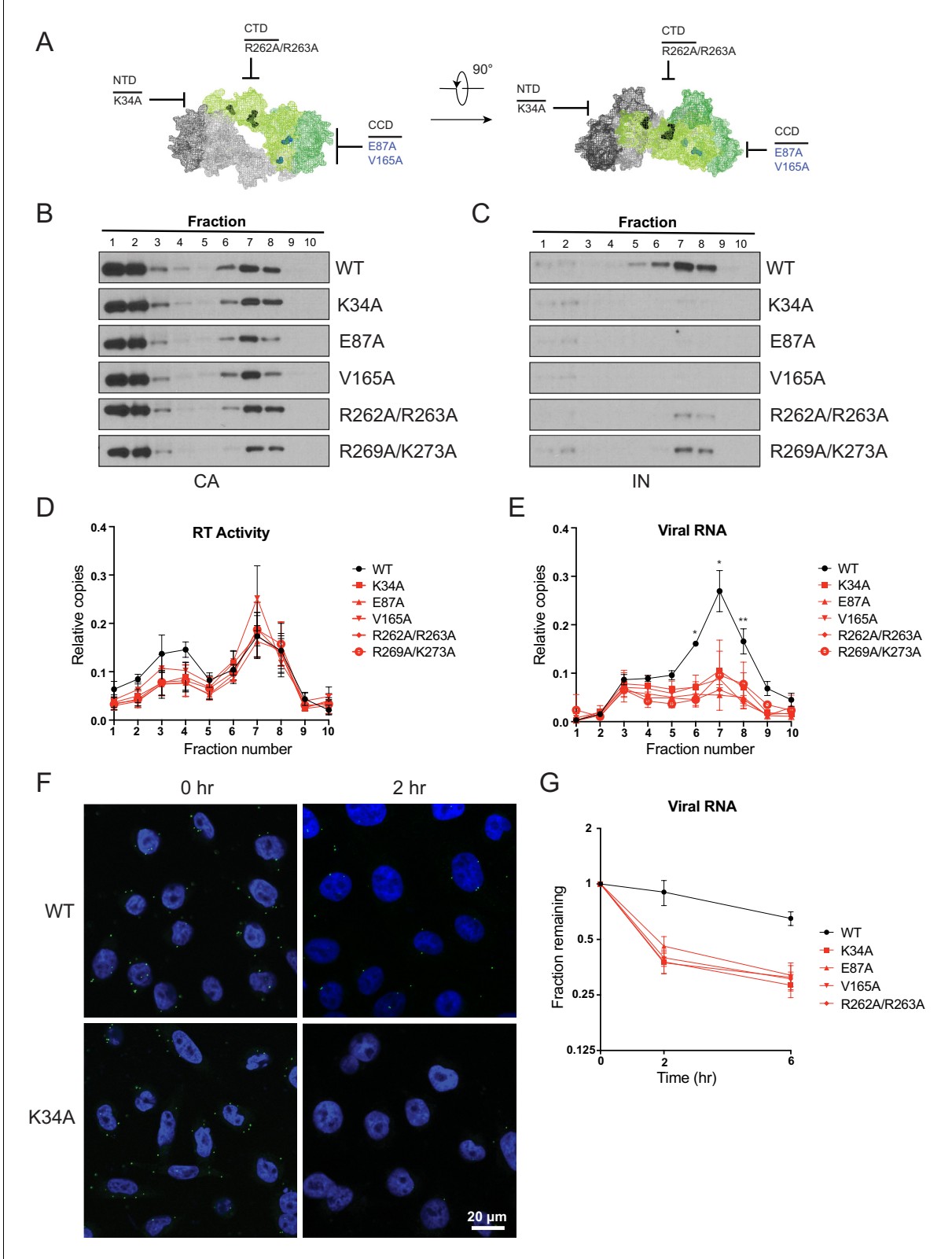

**Figure 8.** Premature loss of vRNA and IN from class II IN mutant viruses upon infection of target cells. (**A**) Locations of the class II IN substitutions K34A, E87A, V165A, and R262A/R263A displayed on a single IN monomer within the context of the HIV-1 IN tetramer intasome structure (PDB 5U1C.) Substitutions are color-coded based on whether they may cause mislocalization of IN in virions (black) or not (blue). (**B–E**) PgsA-745 cells were infected with WT or IN mutant HIV-1 virions and fates of viral core components were analyzed 2 hpi. Fractions were analyzed for the presence of CA (**B**) and IN

*Figure 8 continued on next page*

Figure 8 continued

(C) by immunoblotting and for RT activity (D) and vRNA (E) by Q-PCR. Immunoblots are representative of three independent experiments. Graphed data in (D) and (E) is the average of three independent experiments with error bars indicating standard deviation (*p<0.05 and **p<0.01, by repeated measures one-way ANOVA). (F) Representative images of pgsA-745 cells infected with WT or IN mutant HIV-1$_{NL4-3}$ viruses 0 and 2 hpi. Cells were stained for vRNA (green) and nuclei (blue) as detailed in Materials and Methods. (G) The fraction of viral RNA remaining after 2 and 6 hpi compared to the quantity measured at 0 hpi. MT-4 cells were synchronously infected with VSV-G pseudotyped HIV-1$_{NL4-3}$ viruses and at each time point samples of infected cultures were taken for analysis. Viral RNA levels in samples were measured by Q-PCR and normalized to the levels of GAPDH mRNA. Data points are the average of five independent experiments with error bars indicating standard error of the mean.

The online version of this article includes the following figure supplement(s) for figure 8:

**Figure supplement 1.** Premature loss of vRNA and IN from class II IN mutant viruses upon infection of target cells.

with the IN mutant viruses the reduction in vRNA was greater, and by 2 hr post-infection only 30–40% remained (*Figure 8F*, *Figure 8—figure supplement 1B–C*). These results support the conclusion from the biochemical fractionation experiments that vRNA is prematurely lost from cells infected with class II IN mutant viruses.

Finally, we tested whether our findings were true in physiologically relevant human cells. MT-4 T cells were synchronously infected with WT or class II IN mutant VSV-G pseudotyped HIV-1$_{NL4-3}$ in the presence of nevirapine. Cells were collected immediately after synchronization (0 hr), 2 and 6 hr post-infection, and the quantity of vRNA measured by Q-PCR. In line with the above findings, vRNA levels decreased at a faster rate with the class II IN mutants as compared to WT viruses, with half as much cell-associated vRNA remaining at 2 and 6 hr post-infection for the class II IN mutants (*Figure 8G*). Treating cells with ammonium chloride to prevent fusion of the VSV-G pseudotyped viruses rescued vRNA loss, and vRNA from WT and mutant viruses were retained at equal levels, indicating that the loss of vRNA is dependent on entry into the target cell (*Figure 8—figure supplement 1D*). These findings agree with the previous experiments and demonstrate that class II IN substitutions lead to the premature loss of the vRNA genome also in human T cells.

## Discussion

Our findings highlight the critical role of IN-vRNA interactions in virion morphogenesis and provide the mechanistic basis for how diverse class II IN substitutions lead to similar morphological and reverse transcription defects. We propose that class II IN substitutions lead to the formation of eccentric particles through three distinct mechanisms (*Figure 9*): (i) depletion of IN from virions thus precluding the formation of IN-vRNA complexes; (ii) impairment of functional IN multimerization and as a result, indirect disruption of IN-vRNA binding; and (iii) direct disruption of IN-vRNA binding without substantially affecting IN levels or its inherent multimerization properties. Irrespective of how IN binding to vRNA is inhibited, all substitutions led to the formation of eccentric viruses that were subsequently blocked at reverse transcription in target cells. We provide evidence that premature degradation of the exposed vRNPs and separation of RT from the vRNPs underlies the reverse transcription defect of class II IN mutants (*Figure 8*). Taken together, our findings cement the view that IN binding to RNA accounts for the role of IN in accurate particle maturation and provide the mechanistic basis of why these viruses are blocked at reverse transcription in target cells.

In regard to the first case (i) above, it was previously shown that IN deletion leads to the formation of eccentric particles (*Engelman et al., 1995*; *Jurado et al., 2013*). Thus, it is reasonable to assume that missense mutation that decreased IN levels in virions phenocopy IN deletion viruses. While it is also possible that these substitutions additionally affected IN binding to vRNA or multimerization, we could not reliably address these possibilities due to the extremely low levels of these proteins in virions.

Our results with different oligomeric states of WT IN show that the tetramers rather than dimers or monomers can effectively bind and bridge cognate RNA oligonucleotides (*Figure 5A,B*). Consistent with these observations, a number of substitutions (V165A, K186E, K188E, and K236E) that compromised IN tetramerization also exhibited reduced binding affinity to RNA and failed to bridge between separate RNA oligonucleotides (*Figure 5C*, *Figure 5—figure supplement 1*). More pronounced effects observed with bridging versus binding assays, suggest greater importance of IN tetramerization for forming functionally critical IN-RNA condensates. It is possible that IN dimers can

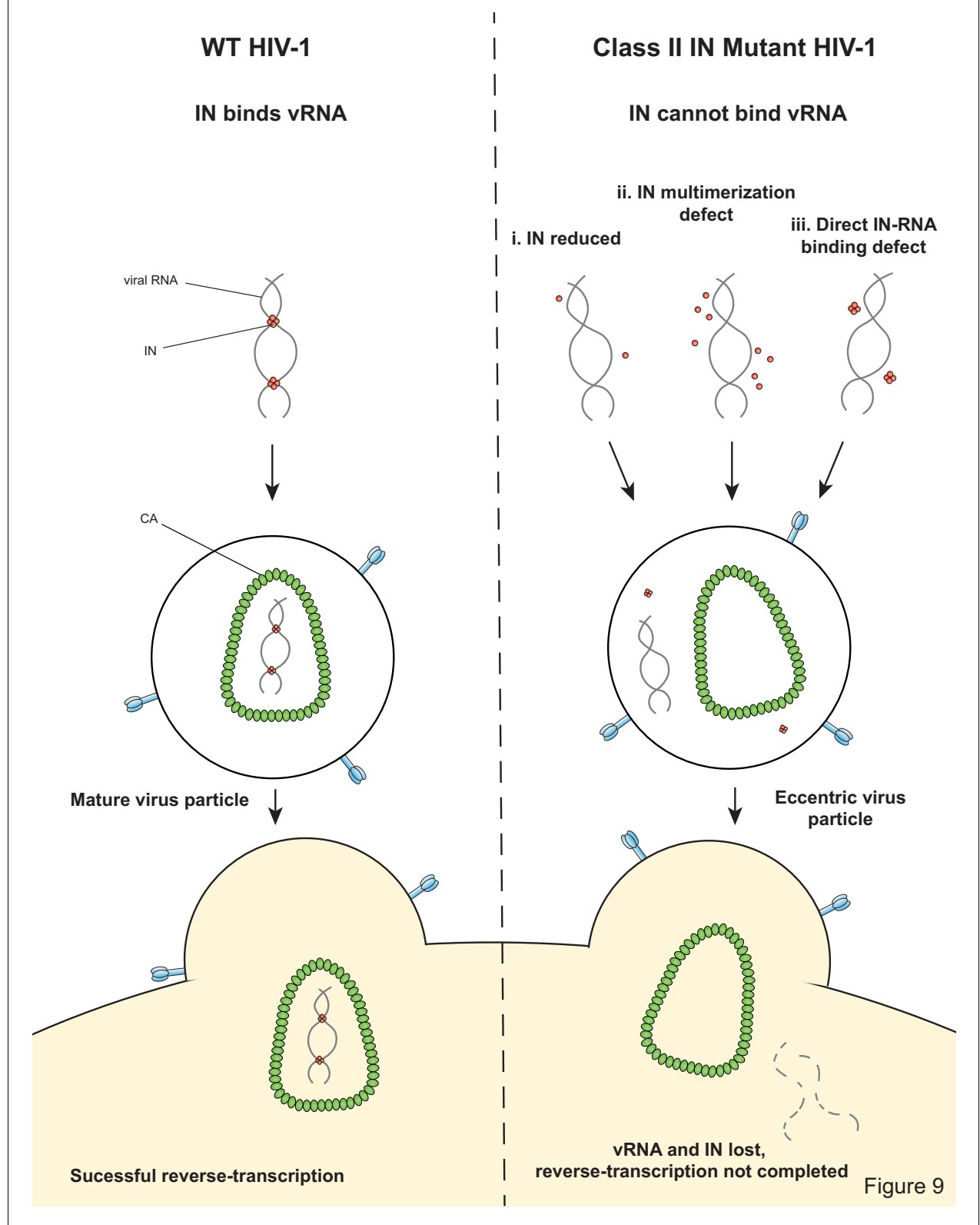

**Figure 9.** Model depicting how class II IN mutants exert their effects on HIV-1 replication.

weakly bind RNA, but they are unable to accommodate and bridge multiple RNA oligonucleotides. By contrast, IN tetramers can both bind with higher affinity and provide a larger binding interface to recruit additional RNA molecules. In turn, the ability of IN tetramers to bridge between different segments of viral RNA could be essential for the formation of stable IN-RNA condensates in virions. Consistent with this notion, all class II substitutions that compromised IN tetramerization also failed to form stable IN-RNA complex in virions (*Figure 3B*). The structural basis for IN binding to RNA is not yet known and it is possible that IN tetramers in the complex with viral RNA differ substantially from those bound to viral DNA within the intasome complex (*Figure 1*). Nonetheless, our findings indicate the importance of IN tetramers for the formation of productive IN-RNA complexes.

Based on MS-based footprinting experiments in vitro, we previously found that positively charged residues within the CTD of IN (i.e. K264, K266, K273) directly contact RNA, as was also validated by CLIP experiments (*Kessl et al., 2016*). Our findings here suggest that IN-vRNA contacts may extend to nearby basic residues within the CTD, such as R262 and R263, and perhaps more surprisingly, K34 within the IN NTD, as alterations of these residues did not prevent IN tetramerization (*Figure 4A–B*) but completely abolished IN-vRNA binding in virions (*Figure 3B*) and reduced RNA-binding in vitro (*Figure 5B–C*). This raises the possibility of a second RNA-binding site in the IN NTD. Structural analysis of IN in complex with RNA will be essential to definitively determine how IN binds RNA as well as the precise multimeric species required for binding and bridging/condensation.

The mechanism by which IN-vRNA interactions mediate the encapsidation of vRNPs inside the CA lattice remains unknown. One possibility is that the temporal coordination of proteolytic cleavage events during maturation is influenced by IN-vRNA interactions (*Könnyű et al., 2013*; *Pettit et al., 2005*). In this scenario, the assembly of the CA lattice may become out of sync with the compaction of vRNA by NC. Another possibility is that IN-vRNA complexes nucleate the assembly of the CA lattice, perhaps by directly binding to CA. Notably, the biochemical assays performed herein show that class II IN substitutions do not appear to affect the assembly and stability of the CA lattice in vitro and in target cells. Although this finding is in disagreement with the previously observed morphological aberrations of the CA lattice present in eccentric particles (*Fontana et al., 2015*), it is possible that the biochemical experiments used herein lack the level of sensitivity required to quantitatively assess these aberrations or that the cryo-ET procedure impacts the structure of the CA lattice in particular in eccentric virions due to the absence of a packaged vRNP complex. A further possibility is that while the CA lattice in class II IN mutant virions may appear morphologically aberrant, they may still uncoat similar to WT virions in target cells. Further studies deciphering the crosstalk between IN-RNA interactions and CA assembly will be critical to our understanding of the role of IN in accurate virion maturation.

While the mislocalization of the vRNA genome in eccentric particles can be accurately assessed by TEM analysis, precisely where IN is located in eccentric particles remains an open question. Earlier studies based on the biochemical separation of core components from detergent-treated IN R269A/K273A virions indicated that IN may also mislocalize outside the CA lattice (*Madison et al., 2017*). In this study, only two class II IN mutants (K34A and R262A/R263A) revealed this phenotype (*Figure 7A,B*). It is intriguing that the bimodal distribution of IN in this experimental setting was only seen with IN mutants that directly inhibited IN binding to vRNA. A possible explanation for these observations is that improperly multimerized IN is retained within the CA lattice or in association with it. Alternatively, class II IN mutants that may localize outside of the CA lattice form dense aggregates and as a result migrate similar to WT IN. The fact that select class II IN mutants migrate similar to WT IN upon CA destabilization (*Figure 7—figure supplement 1A*) but are rapidly lost in infected cells (*Figure 8C*), strongly suggests the latter possibility.

Why is the unprotected vRNA and IN prematurely lost in target cells? It seems evident that the protection afforded by the CA lattice matters the most for vRNP stability, though we cannot rule out that IN binding to vRNA may in and of itself stabilize both the genome and IN. Alternatively, the AU-rich nucleotide content of HIV-1 may destabilize its RNA (*Maldarelli et al., 1991*; *Schwartz et al., 1992a*; *Schwartz et al., 1992b*), similar to several cellular mRNAs that encode for cytokines and growth factors (*Wu and Brewer, 2012*). Finally, RNA nicking and deadenylation in virions by virion-associated enzymes (*Gorelick et al., 1999*; *Miyazaki et al., 2010*; *Sakuragi et al., 2001*) may predispose retroviral genomes to degradation when they are prematurely exposed to the cytosolic milieu. While cytosolic IN undergoes proteasomal degradation when expressed alone

in cells (*Mulder and Muesing, 2000*; *Ali et al., 2019*; *Llano et al., 2004*; *Zheng et al., 2011*; *Devroe et al., 2003*), we have found previously that proteasome inhibition does not rescue vRNA or IN in target cells infected with a class II IN mutant (*Madison et al., 2017*). Further studies are needed to determine whether a specific cellular mechanism or an inherent instability of vRNPs is responsible for the loss of vRNA in infected cells.

Although the basic aspects of virion maturation are conserved, particle morphologies are vastly different across retroviruses (*Martin et al., 2016*; *Zhang et al., 2015*). It is currently unknown whether IN molecules of other retroviruses regulate viral maturation through binding to vRNAs. Interestingly, mutations within the C-terminus of murine leukemia virus (MLV) IN can similarly cause defects in reverse transcription (*Steinrigl et al., 2007*; *Lai et al., 2001*), raising the possibility of a conserved role for IN in particle maturation and reverse transcription across retroviruses.

In conclusion, we have identified IN-vRNA binding as the underlying factor for the role of IN in virion morphogenesis and show that virion morphogenesis is necessary to prevent the premature loss of vRNA and IN early in the HIV-1 lifecycle. Despite relatively high barriers, drugs that inhibit the catalytic activity of IN do select for resistance, and additional drug classes that inhibit IN activity through novel mechanisms of action would be a valuable addition to currently available treatments. The finding that IN-vRNA interaction can be inhibited in multiple ways – by directly altering residues in the IN CTD or by altering IN multimerization in virions – can help guide the design of future anti-retroviral compounds.

# Materials and methods

## Key resources table

| Reagent type (species) or resource | Designation | Source or reference | Identifiers | Additional information |
|---|---|---|---|---|
| Gene (human immunodeficiency virus type 1) | Integrase (IN) | NCBI (NC_001802.1) | Gene ID: 155348 | |
| Strain, strain background (*Escherichia coli*) | DH10B | Thermo Fisher Scientific | EC0113 | Competent cells |
| Strain, strain background (*Escherichia coli*) | BL21 | Thermo Fisher Scientific | C600003 | Competent cells |
| Cell line (*Homo sapiens*) | HEK293T | ATCC | CRL-11268 | |
| Cell line (*Homo sapiens*) | TZM-bl | NIH AIDS Reagent Program | 8129 | |
| Cell line (*Cricetulus griseus*) | pgsA-745 | ATCC | CRL-2242 | Xylosyltransferase I deficient |
| Cell line (*Homo sapiens*) | MT-4 | NIH AIDS Reagent Program | 120 | |
| Antibody | Anti-HIV-1 integrase-4 (mouse monoclonal) | *Bouyac-Bertoia et al., 2001* | | WB (1:4000), IP (5 μL/100 μL of Dyna beads) |
| Antibody | Anti-HIV-1 integrase-1 (rabbit polyclonal) | This paper | | WB (1:1000) |
| Antibody | Anti-HIV-1 p24 antibody (mouse monoclonal) | NIH AIDS Reagent Program | 183-H12-5C | WB (1:100) |

*Continued on next page*

*Continued*

| Reagent type (species) or resource | Designation | Source or reference | Identifiers | Additional information |
|---|---|---|---|---|
| Antibody | Anti-HIV-1 reverse transcriptase antibody (rabbit polyclonal) | NIH AIDS Reagent Program | 6195 | WB (1:1000) |
| Antibody | Anti-HIV-1 Vpr antibody (rabbit polyclonal) | NIH AIDS Reagent Program | 11836 | WB (1:1000) |
| Antibody | Anti-HIV-1 MA antibody (rabbit polyclonal) | NIH AIDS Reagent Program | 4811 | WB (1:1000) |
| Commercial assay, kit | QuikChange Site-Directed Mutagenesis Kit | Agilent Technologies | Cat# 200519 | |
| Commercial assay, kit | DNeasy Blood and Tissue kit | Qiagen | Cat# 69506 | |
| Commercial assay, kit | RNAScope F luorescent Multiplex Detection Reagents | Advanced Cell Diagnostics | Ref# 320851 | |
| Chemical compound, drug | Nevirapine | NIH AIDS Reagents | | |
| Chemical compound, drug | Ethylene glycol bis(succinimidyl succinate) (EGS) | Thermo Fisher Scientific | | |

## Plasmids

The pNLGP plasmid consisting of the HIV-1$_{NL4-3}$-derived Gag-Pol sequence inserted into the pCR/V1 plasmid backbone (*Zennou et al., 2004*) and the CCGW vector genome plasmid carrying a GFP reporter under the control of the CMV promoter (*Cowan et al., 2002*; *Hatziioannou et al., 2003*) were previously described. The pLR2P-vprIN plasmid expressing a Vpr-IN fusion protein has also been previously described (*Liu et al., 1997*). Mutations in the IN coding sequence were introduced into both the pNLGP plasmid and the HIV-1$_{NL4-3}$ full-length proviral plasmid (pNL4-3) by overlap extension PCR. Briefly, forward and reverse primers containing IN mutations in the *pol* reading frame were used in PCR reactions with antisense and sense outer primers containing unique restriction endonuclease sites (AgeI-sense, NotI-antisense for NLGP and AgeI-sense, EcoRI-antisense for pNL4-3), respectively. The resulting fragments containing the desired mutations were mixed at a 1:1 ratio and overlapped subsequently using the sense and antisense primer pairs. The resulting fragments were digested with the corresponding restriction endonucleases and cloned into pNLGP and pNL4-3 plasmids. IN mutations were introduced into the pLR2P-vprIN plasmid using the QuickChange Site-Directed Mutagenesis kit (Agilent Technologies). The presence of the desired mutations and the absence of unwanted secondary changes were verified by Sanger sequencing. HIV-1 CA K203A substitution, which destabilizes the CA lattice (*Forshey et al., 2002*), was cloned into HIV-1$_{NL4-3}$ bearing class II IN mutations by conventional cloning. Generation of Gag-bZIP chimeras bearing the leucine zipper domain from the yeast GCN4 protein in place of NC, which facilitates Gag dimerization but does not bind RNA has been described previously (*Johnson et al., 2002*; *Accola et al., 2000*). A version of this chimera was generated by replacing NC with a NotI restriction site and cloning of the PCR-amplified GCN4-bZIP in its place. Class II IN mutants were subsequently cloned into this backbone by the QuickChange Site-Directed Mutagenesis kit (Agilent Technologies).

## Cells and viruses

All cell lines were originally obtained from American Type Culture Collection and NIH AIDS Reagents where STR profiling was performed. MT-4 cells were additionally subjected to STR profiling at Washington University School of Medicine Genome Engineering and iPSC center. All cell lines are regularly checked for mycoplasma contamination using the MycoAlert mycoplasma detection kit (Lonza) and verified to be free of contamination during these studies. HEK293T cells (ATCC CRL-11268) and HeLa-derived TZM-bl cells (NIH AIDS Reagent Program) were maintained in Dulbecco's modified Eagle's medium supplemented with 10% fetal bovine serum. MT-4 cells were maintained in RPMI 1640 medium supplemented with 10% fetal bovine serum. CHO K1-derived pgsA-745 cells (CRL-2242, ATCC) that lack a functional xylosyltransferase enzyme and as a result do not produce glycosaminoglycans were maintained in Dulbecco's modified Eagle's/F12 (1:1) media supplemented with 10% fetal bovine serum and 1 mM L-glutamine. Single-cycle GFP reporter viruses pseudotyped with vesicular stomatitis virus G protein (VSV-G) were produced by transfection of HEK293T cells with pNLGP-derived plasmids, the CCGW vector genome carrying GFP, and VSV-G expression plasmid at a ratio of 5:5:1, respectively, using polyethyleneimine (PolySciences, Warrington, PA). Full-length viruses pseudotyped with VSV-G were produced by transfecting HEK293T cells with the pNL4-3-derived plasmids and VSV-G plasmid at a ratio of 4:1 (pNL4-3:VSV-G).

## Immunoblotting

Viral and cell lysates were resuspended in sodium dodecyl sulfate (SDS) sample buffer and separated by electrophoresis on Bolt 4–12% Bis-Tris Plus gels (Life Technologies), blotted onto nitrocellulose membranes and probed overnight at 4°C with the following antibodies in Odyssey Blocking Buffer (LI-COR): mouse monoclonal anti-HIV p24 antibody (183-H12-5C, NIH AIDS reagents), mouse monoclonal anti-HIV integrase antibody (*Bouyac-Bertoia et al., 2001*), rabbit polyclonal anti-HIV integrase antibody raised in-house against Q44-LKGEAMHGQVD-C56 peptide and hence unlikely to be affected by the substitutions introduced into IN in this study, rabbit polyclonal anti-HIV-1 reverse transcriptase antibody (6195, NIH AIDS reagents), rabbit polyclonal anti-Vpr antibody (11836, NIH AIDS Reagents), rabbit polyclonal anti-MA antibody (4811, NIH AIDS Reagents). Membranes were probed with fluorophore-conjugated secondary antibodies (LI-COR) and scanned using an LI-COR Odyssey system. IN and CA levels in virions were quantified using Image Studio software (LI-COR). For analysis of the fates of core components in infected cells, antibody incubations were done using 5% non-fat dry milk. Membranes were probed with HRP-conjugated secondary antibodies and developed using SuperSignal West Femto reagent (Thermo-Fisher).

## Analysis of reverse transcription products in infected cells

MT-4 cells were grown in 24-well plates and infected with VSV-G pseudotyped pNL4-3 viruses (either WT or class II IN mutant) at a multiplicity of infection (MOI) of 2 in the presence of polybrene. After 6 hr, post-infection cells were collected, pelleted by brief centrifugation, and resuspended in PBS. DNA was extracted from cells using the DNeasy Blood and Tissue Kit (Qiagen) as per kit protocol. Quantity of HIV-1 vDNA was measured by Q-PCR using primers specific for early reverse-transcripts.

## Vpr-IN transcomplementation experiments

A class I IN mutant virus (HIV-1$_{NL4-3}$ IN$_{D116N}$) was trans-complemented with class II mutant IN proteins as described previously (*Liu et al., 1997*). Briefly, HEK293T cells grown in 24-well plates were co-transfected with a derivative of the full-length HIV-1$_{NL4-3}$ proviral plasmid bearing a class I IN substitution (pNL4-3$_{D116N}$), VSV-G, and derivatives of the pLR2P-vprIN plasmid bearing class II IN mutations at a ratio of 6:1:3. Two days post-transfection cell-free virions were collected from cell culture supernatants. Integration capability of the trans-complemented class II IN mutants was tested by infecting MT-4 cells and measuring the yield of progeny virions in cell culture supernatants over a 6 d period as described previously (*Liu et al., 1997*). In brief, MT-4 cells were incubated with virus inoculum in 96 V-bottom well plates for 4 hr at 37°C after which the virus inoculum was washed away and replaced with fresh media. Immediately following removal of the virus inoculum and during the 6 subsequent days, the number of virions present in the culture supernatant was quantified by measuring RT activity using a Q-PCR-based assay (*Pizzato et al., 2009*).

## CLIP experiments

CLIP experiments were conducted as previously described (*Kessl et al., 2016*; *Kutluay et al., 2014*; *Kutluay and Bieniasz, 2016*). Cell-free HIV-1 virions were isolated from transfected HEK293T cells. Briefly, cells in 15 cm cell culture plates were transfected with 30 µg full-length proviral plasmid (pNL4-3) DNA containing the WT sequence or indicated *pol* mutations within the IN coding sequence. Cells were grown in the presence of 4-thiouridine for 16 hr before the virus harvest. Two days post-transfection cell culture supernatants were collected and filtered through 0.22 µm filters and pelleted by ultracentrifugation through a 20% sucrose cushion using a Beckman SW32-Ti rotor at 28,000 rpm for 1.5 hr at 4°C. Virus pellets were resuspended in phosphate-buffered saline (PBS) and UV-crosslinked. Following lysis in RIPA buffer, IN-RNA complexes were immunoprecipitated using a mouse monoclonal anti-IN antibody (*Bouyac-Bertoia et al., 2001*). Bound RNA was end-labeled with $\gamma$-$^{32}$P-ATP and T4 polynucleotide kinase. The isolated protein-RNA complexes were separated by SDS-PAGE, transferred to nitrocellulose membranes, and exposed to autoradiography films to visualize RNA. Lysates and immunoprecipitates were also analyzed by immunoblotting using antibodies against IN.

## IN multimerization in virions

HEK293T cells grown on 10 cm dishes were transfected with 10 µg pNL4-3 plasmid DNA containing the WT sequence or indicated *pol* mutations within IN coding sequence. Two days post-transfection cell-free virions collected from cell culture supernatants were pelleted by ultracentrifugation through a 20% sucrose cushion using a Beckman SW41-Ti rotor at 28,000 rpm for 1.5 hr at 4°C. Pelleted virions were resuspended in 1× PBS and treated with EGS (ThermoFisher Scientific), a membrane-permeable crosslinker, at a concentration of 1 mM for 30 min at room temperature. Crosslinking was stopped by the addition of SDS sample buffer. Samples were subsequently separated on 3–8% Tris-acetate gels and analyzed by immunoblotting using a mouse monoclonal anti-IN antibody (*Bouyac-Bertoia et al., 2001*).

## Size exclusion chromatography (SEC)

All of the mutations were introduced into a plasmid backbone expressing His$_6$ tagged pNL4-3-derived IN by QuikChange site-directed mutagenesis kit (Agilent) (*Kessl et al., 2012*). His$_6$ tagged recombinant pNL4-3 WT and mutant INs were expressed in BL21 (DE3) *E. coli* cells followed by nickel and heparin column purification as described previously (*Kessl et al., 2012*; *Cherepanov, 2007*). Recombinant WT and mutant INs were analyzed on Superdex 200 10/300 GL column (GE Healthcare) with running buffer containing 20 mM HEPES (pH 7.5), 1 M NaCl, 10% glycerol, and 5 mM BME at 0.3 mL/min flow rate. The proteins were diluted to 10 µM with the running buffer and incubated for 1 hr at 4°C followed by centrifugation at 10,000 g for 10 min. Multimeric form determination was based on the standards including bovine thyroglobulin (670,000 Da), bovine gamma-globulin (158,000 Da), chicken ovalbumin (44,000 Da), horse myoglobin (17,000 Da), and vitamin B12 (1350 Da).

## Analysis of IN-RNA binding in vitro

Following SEC of IN as above, individual fractions of tetramer, dimer, and monomer forms were collected and their binding to TAR RNA was analyzed by an Alpha screen assay as described previously (*Kessl et al., 2016*). Briefly, 100 nM His$_6$ tagged IN fractions (tetramer, dimer, and monomer) were incubated with nickel acceptor beads while increasing concentrations of biotinylated-TAR RNA was incubated with streptavidin donor beads in buffer containing 100 mM NaCl, 1 mM MgCl$_2$, 1 mM DTT, 1 mg/mL BSA, and 25 mM Tris (pH 7.4). Followed by 2 hr incubation at 4°C, they were mixed and the reading was taken after 1 hr incubation at 4°C by PerkinElmer Life Sciences Enspire multi-mode plate reader. The Kd values were calculated using OriginLab software.

## AlphaScreen-based RNA bridging assays

The RNA bridging property of IN was analyzed by AlphaScreen-based assay as described (*Kessl et al., 2016*). Briefly, equal concentrations (1 nM) of two synthetic TAR RNA oligonucleotides labeled either with biotin or DIG were mixed and then streptavidin donor and anti-DIG acceptor beads at 0.02 mg/mL concentration were supplied in a buffer containing 100 mM NaCl, 1 mM

MgCl$_2$, 1 mM DTT, 1 mg/mL BSA, and 25 mM Tris (pH 7.4). After 2 hr incubation at 4°C, indicated concentrations of IN were added to the reaction mixture and incubated further for 1.5 hr at 4°C. AlphaScreen signals were recorded with a PerkinElmer Life Sciences Enspire multimode plate reader.

## Virus production and transmission electron microscopy

Cell-free HIV-1 virions were isolated from transfected HEK293T cells. Briefly, cells grown in two 15 cm cell culture plates (10$^7$ cells per dish) were transfected with 30 µg full-length proviral plasmid (pNL4-3) DNA containing the WT sequence or indicated *pol* mutations within IN coding sequence using PolyJet DNA transfection reagent as recommended by the manufacturer (SignaGen Laboratories). Two days after transfection, cell culture supernatants were filtered through 0.22 µm filters, and pelleted by ultracentrifugation using a Beckman SW32-Ti rotor at 26,000 rpm for 2 hr at 4°C. Fixative (2.5% glutaraldehyde, 1.25% paraformaldehyde, 0.03% picric acid, 0.1 M sodium cacodylate, pH 7.4) was gently added to resulting pellets, and samples were incubated overnight at 4°C. The following steps were conducted at the Harvard Medical School Electron Microscopy core facility. Samples were washed with 0.1 M sodium cacodylate, pH 7.4 and postfixed with 1% osmium tetroxide/1.5% potassium ferrocyanide for 1 hr, washed twice with water, once with maleate buffer (MB), and incubated in 1% uranyl acetate in MB for 1 hr. Samples washed twice with water were dehydrated in ethanol by subsequent 10 min incubations with 50%, 70%, 90%, and then twice with 100%. The samples were then placed in propyleneoxide for 1 hr and infiltrated overnight in a 1:1 mixture of propyleneoxide and TAAB Epon (Marivac Canada Inc). The following day the samples were embedded in TAAB Epon and polymerized at 60°C for 48 hr. Ultrathin sections (about 60 nm) were cut on a Reichert Ultracut-S microtome, transferred to copper grids stained with lead citrate, and examined in a JEOL 1200EX transmission electron microscope with images recorded on an AMT 2 k CCD camera. Images were captured at 30,000× magnification, and over 100 viral particles per sample were counted by visual inspection.

## Equilibrium density sedimentation of virion core components in vitro

Equilibrium density sedimentation of virion core components was performed as previously described (*Madison et al., 2017*). Briefly, HEK293T cells grown in 10 cm cell culture plates were transfected with 10 µg pNLGP plasmid DNA containing the WT sequence or indicated *pol* mutations within IN coding sequence. Two days post-transfection cell-free virions collected from cell culture supernatants were pelleted by ultracentrifugation through a 20% sucrose cushion using a Beckman SW41-Ti rotor at 28,000 rpm for 1.5 hr at 4°C. Pelleted viral-like particles were resuspended in PBS and treated with 0.5% Triton X-100 for 2 min at room temperature. Immediately after, samples were layered on top of 30–70% linear sucrose gradients prepared in 1× STE buffer (100 mM NaCl, 10 mM Tris-Cl [pH 8.0], 1 mM EDTA) and ultracentrifuged using a Beckman SW55-Ti rotor at 28,500 rpm for 16 hr at 4°C. Fractions (500 µL) collected from the top of the gradients were analyzed for IN, CA, and MA by immunoblotting as detailed above.

## Biochemical analysis of virion core components in infected cells

Biochemical analysis of retroviral cores in infected cells was performed as described previously (*Kutluay et al., 2013*). Briefly, pgsA-745 cells were infected with VSV-G pseudotyped single cycle GFP-reporter viruses or its derivatives synchronously at 4°C. Following the removal of virus inoculum and extensive washes with PBS, cells were incubated at 37°C for 2 hr. To prevent loss of vRNA due to reverse-transcription, cells were infected in the presence of 25 µM nevirapine. Post-nuclear supernatants were separated by ultracentrifugation on 10–50% linear sucrose gradients using a Beckman SW55-Ti rotor at 30,000 rpm for 1 hr at 4°C. Ten 500 µL fractions from the top of the gradient were collected, and CA, IN, and vRNA in each fraction were analyzed by either immunoblotting or Q-PCR (*Kutluay et al., 2013*). A SYBR-Green-based Q-PCR assay (*Pizzato et al., 2009*) was used to determine RT activity in the collected sucrose fractions.

## Visualization of vRNA in infected cells

Viral RNA was visualized in infected cells according to the published multiplex immunofluorescent cell-based detection of DNA, RNA, and Protein (MICDDRP) protocol (*Puray-Chavez et al., 2017*).

VSV-G pseudotyped HIV-1$_{NL4-3}$ virus stocks were prepared as described above and concentrated 40X using a lentivirus precipitation solution (ALSTEM). PgsA-745 cells were plated on 1.5 mm collagen-treated coverslips (GG-12–1.5-Collagen, Neuvitro) placed in 24-well plates 1 d before infection. Synchronized infections were performed by incubating pre-chilled virus inoculum on the cells for 30 min at 4°C. Cells were infected with WT virus at an MOI of 0.5, or with an equivalent number (normalized by RNA copy number) of IN mutant viral particles. After removal of the virus, inoculum cells were washed with PBS and either immediately fixed with 4% paraformaldehyde, or incubated at 37°C for 2 hr before fixing. To prevent loss of vRNA due to reverse-transcription, cells were infected and incubated in the presence of 25 µM nevirapine. Following fixation, cells were dehydrated with ethanol and stored at −20°C. Before probing for vRNA, cells were rehydrated, incubated in 0.1% Tween in PBS for 10 min, and mounted on slides. Probing was performed using RNAScope probes and reagents (Advanced Cell Diagnostics). Briefly, coverslips were treated with protease solution for 15 min in a humidified HybEZ oven (Advanced Cell Diagnostics) at 40°C. The coverslips were then washed with PBS and pre-designed anti-sense probes (Puray-Chavez et al., 2017) specific for HIV-1 vRNA were applied and allowed to hybridize with the samples in a humidified HybEZ oven at 40°C for 2 hr. The probes were visualized by hybridizing with preamplifiers, amplifiers, and finally, a fluorescent label. First, pre-amplifier 1 (Amp 1-FL) was hybridized to its cognate probe for 30 min in a humidified HybEZ oven at 40°C. Samples were then subsequently incubated with Amp 2-FL, Amp 3-FL, and Amp 4A-FL for 15 min, 30 min, and 15 min, respectively. Between adding amplifiers, the coverslips were washed with a proprietary wash buffer. Nuclei were stained with DAPI diluted in PBS at room temperature for 5 min. Finally, coverslips were washed in PBST followed by PBS and then mounted on slides using Prolong Gold Antifade.

## Microscopy and image quantification

Images were taken using a Zeiss LSM 880 Airyscan confocal microscope equipped with a × 63/1.4 oil-immersion objective using the Airyscan super-resolution mode. Ten images were taken for each sample using the × 63 objective. Numbers of nuclei and vRNA punctae in images were counted using Volocity software (Quorum Technologies). The number of vRNA punctae per 100 nuclei were recorded at 0 hr post-infection (hpi) and 2 hpi for each virus, and the number at 2 hpi compared to the number at 0 hpi.

## Analysis of the fate of vRNA genome in MT4 cells

MT-4 cells were infected with VSV-G pseudotyped HIV-1 NL4-3 WT or an equivalent number of mutant viruses (normalized by RT activity) synchronously at 4°C. After removal of virus inoculum and extensive washes with PBS, cells were incubated at 37°C for 6 hr in the presence of 25 µM nevirapine. Immediately after synchronization (0 hr) and at 2 and 6 hr post-infection samples were taken from the infected cultures and RNA was isolated using TRIzol Reagent. The amount of vRNA was measured by Q-RT-PCR.

For each experiment, IN signal was normalized to CA signal for each virus, and the resulting value compared to that of WT (set at 100%). Reported values are the average value (as a percent of WT) and standard deviation (SD) between four independent experiments. Mutants with less than 20% IN signal of WT are highlighted in gray.

## Acknowledgements

We thank Dr. Michael Malim for providing the anti-IN monoclonal antibody and members of the Tolia lab for assisting in PyMol analysis. This study was supported by grants NIH grants P50 GM103297 (the Center for HIV RNA Studies) and GM122458 to SBK, AI143389-F31 fellowship to JE, R01 AI062520 to MK and SBK, U54 AI150472 to MK and AE, AI070042 to AE.

## Additional information

### Funding

| Funder | Grant reference number | Author |
|---|---|---|
| National Institutes of Health | GM103297 | Sebla B Kutluay |
| National Institutes of Health | GM122458 | Sebla B Kutluay |
| National Institutes of Health | AI143389 | Jennifer L Elliott |
| National Institutes of Health | AI062520 | Mamuka Kvaratskhelia<br>Sebla B Kutluay |
| National Institutes of Health | AI150472 | Mamuka Kvaratskhelia |
| National Institutes of Health | AI070042 | Alan N Engelman |

The funders had no role in study design, data collection and interpretation, or the decision to submit the work for publication.

### Author contributions

Jennifer L Elliott, Conceptualization, Data curation, Formal analysis, Funding acquisition, Validation, Investigation, Visualization, Methodology, Writing - original draft, Writing - review and editing; Jenna E Eschbach, Data curation, Formal analysis, Validation, Investigation, Visualization, Methodology; Pratibha C Koneru, Wen Li, Conceptualization, Data curation, Formal analysis, Validation, Investigation, Visualization, Methodology; Maritza Puray-Chavez, Conceptualization, Resources, Supervision, Methodology; Dana Townsend, Data curation, Formal analysis, Investigation; Dana Q Lawson, Formal analysis, Investigation, Methodology; Alan N Engelman, Mamuka Kvaratskhelia, Conceptualization, Resources, Supervision, Funding acquisition, Investigation, Methodology, Writing - original draft, Project administration, Writing - review and editing; Sebla B Kutluay, Conceptualization, Resources, Data curation, Formal analysis, Supervision, Funding acquisition, Validation, Investigation, Methodology, Writing - original draft, Project administration, Writing - review and editing

### Author ORCIDs

Pratibha C Koneru (iD) https://orcid.org/0000-0003-3955-4548
Mamuka Kvaratskhelia (iD) http://orcid.org/0000-0003-3800-0033
Sebla B Kutluay (iD) https://orcid.org/0000-0001-5549-7032

### Decision letter and Author response

Decision letter https://doi.org/10.7554/eLife.54311.sa1
Author response https://doi.org/10.7554/eLife.54311.sa2

## Additional files

### Supplementary files

- Supplementary file 1. Quantitation of IN in virions as measured by western blotting.

- Supplementary file 2. Predominant multimeric species of mutant INs in vitro as assessed by SEC.

- Transparent reporting form

### Data availability

All data generated or analysed during this study are included in the manuscript and supporting files.

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
