## [Decision Letter]

**Acceptance summary:**

The results presented in this manuscript point to an essential role for integrase/RNA interactions during maturation of virions. The experimental data is solid and clearly presented.

**Decision letter after peer review:**

Thank you for submitting your article "Integrase-RNA interactions underscore the critical role of integrase in HIV-1 virion morphogenesis" for consideration by *eLife*. Your article has been reviewed by Päivi Ojala as the Senior Editor, a Reviewing Editor, and three reviewers. The following individual involved in review of your submission has agreed to reveal their identity: Eric Freed (Reviewer #2).

The reviewers have discussed the reviews with one another and the Reviewing Editor has drafted this decision to help you prepare a revised submission.

Summary:

In this study, Elliott and colleagues investigate the properties of "class II" HIV-1 integrase (IN) mutants. These class II HIV-1 integrase mutants do not compromise the catalytic activity of the enzyme but severely impair viral replication. The underlying mechanisms are not well understood, but aberrant virion morphology is a hallmark of this class of mutant.

The authors suggest that these mutants interfere with binding between IN and the viral RNA by one of three mechanisms: (1) reducing IN levels in virions, (2) inhibiting IN multimerization, and/or (3) directly disrupting IN-RNA interactions. The results point to an essential role for integrase/RNA interactions during maturation of virions.

This study provides significant new insights into the properties of class II IN mutants, which have for long been of interest and display properties that are mirrored by allosteric IN inhibitors. The experimental data is solid and clearly presented.

Essential revisions:

The authors analyze a diverse collection of previously obscure class II IN mutants and provide evidence that many of them are defective for the formation of tetramers and that it is the tetrameric form that binds viral RNA. Their data seem to clarify the location of defective IN, which they show is associated with HIV-1 cores (biochemical fractionation, Figure 6), while the RNA genome is outside (according to EM, Figure 5). It was refreshing to see all these mutants, including δ-IN and H12N, analyzed using several complementary approaches side-by-side. However, a few issues outlined below need to be carefully addressed.

1) The conclusion that RNA binding is exclusive to the tetrameric form of IN is very important in this work, and it does make sense. Please add more evidence that the HIV needs IN tetramer to bind RNA. In the current manuscript, this conclusion is based on a single experiment (Figure 4C): WT IN was separated into tetramer, dimer and a monomer by SEC, and the three fractions are tested for the ability to bind RNA. Are these forms really not supposed to interconvert? For example, what if dimers do bind RNA, but they fall apart during chromatography and/or downstream processing? Would it not be better to test RNA binding with class II mutants that fail to form tetramers (Figure 4B: K188E, K236E)? In addition, there are HIV-1 IN mutants that do not form tetramers as recombinant proteins (K186E or Y15A, for example); would these not bind RNA? It would be essential to expand Figure 4D including mutations of residues that are not expected to interact with vRNA directly. Because IN mutants stick with cores whilst vRNA is not (Figure 6 and Figure 5), they may be physically separated within virions. The CLIP data (Figure 3B) is consistent with this and could be a consequence of the physical separation. This is another reason to test RNA binding by these mutants directly.

The crosslinking results in Figure 4A do not seem very convincing: there are high-MW IN species visible for all mutants. Perhaps some of them are hetero-multimer crosslinks (i.e. with other proteins in the virion?). SEC data, such as in panel B, are much clearer; I would suggest including more mutants in this analysis.

Moreover, it is not clear how reproducible the cross-linking patterns are (and the WT in the third panel looks very different from the WTs in the first and second panels). The cross-linking results, to the extent that they can be interpreted, appear to be very different from the SEC profiles shown in Figure 4B. The authors conclude (subsection “IN multimerization plays a key role in RNA binding”) that the majority of class II IN mutants are predominantly monomeric in virions, yet the majority of the protein is found in high-MW aggregates. This issue needs to be addressed.

2) The key issue of whether the IN mutants are excluded or not from viral cores needs to be clarified. The sedimentation of the majority of IN mutants in the dense, CA-containing fractions in Figure 6A would seem to suggest that these mutant IN proteins are located within the viral capsid. However, in the cell-based uncoating assay, the IN mutants are degraded (Figure 7C) while a high percentage of capsids remain intact at the 2 hour time point (Figure 7B). These data suggest that the mutant IN proteins are excluded from the protective environment of the capsid and are thus rapidly degraded. Indeed, in the model figure (Figure 8), these mutants are depicted as being outside the cores. One possible solution to this conundrum is that the IN mutants are prone to aggregation under the conditions used in Figure 6 and thus co-sediment with CA without actually being inside the capsid. In any case, as currently presented, the data appear to be in contradiction and leave this important question unresolved.

The data presented in Figure 8 suggest that the class II IN mutations do not affect the assembly or stability of the CA lattice either in vitro or in target cells (but see Fontana et al., 2015). Yet, in the Figure 8 model, the capsid is shown as discontinuous or "broken" in the context of the IN mutants, a feature that would presumably significantly compromise capsid stability. The model thus appears to be inconsistent with the data.

3) The difference between tetramers, dimers and monomers in Figure 4C is surprising. The species would be expected to re-equilibrate after SEC. Please comment.

---

## [Author Response]

Essential revisions:The authors analyze a diverse collection of previously obscure class II IN mutants and provide evidence that many of them are defective for the formation of tetramers and that it is the tetrameric form that binds viral RNA. Their data seem to clarify the location of defective IN, which they show is associated with HIV-1 cores (biochemical fractionation, Figure 6), while the RNA genome is outside (according to EM, Figure 5). It was refreshing to see all these mutants, including δ-IN and H12N, analyzed using several complementary approaches side-by-side. However, a few issues outlined below need to be carefully addressed.1) The conclusion that RNA binding is exclusive to the tetrameric form of IN is very important in this work, and it does make sense. Please add more evidence that the HIV needs IN tetramer to bind RNA. In the current manuscript, this conclusion is based on a single experiment (Figure 4C): WT IN was separated into tetramer, dimer and a monomer by SEC, and the three fractions are tested for the ability to bind RNA. Are these forms really not supposed to interconvert? For example, what if dimers do bind RNA, but they fall apart during chromatography and/or downstream processing? Would it not be better to test RNA binding with class II mutants that fail to form tetramers (Figure 4B: K188E, K236E)? In addition, there are HIV-1 IN mutants that do not form tetramers as recombinant proteins (K186E or Y15A, for example); would these not bind RNA? It would be essential to expand Figure 4D including mutations of residues that are not expected to interact with vRNA directly. Because IN mutants stick with cores whilst vRNA is not (Figure 6 and Figure 5), they may be physically separated within virions. The CLIP data (Figure 3B) is consistent with this and could be a consequence of the physical separation. This is another reason to test RNA binding by these mutants directly.

We thank the reviewer for the insightful comments. As suggested, we have determined the RNA binding properties of additional mutants that do not form tetramers in vitro (K186E, K188E and K236E) alongside with K34A, R262A/R263A and the canonical class II IN mutant V165A using an RNA binding and RNA-bridging assay (Kessl et al., 2016) as shown in Figure 5 and Figure 5—figure supplement 1 of the current manuscript. The latter approach relies on IN-mediated bridging of fluorescently labeled target RNA oligonucleotides (i.e. TAR) and as such is a good proxy for assessing the multimerization-dependent RNA binding ability of IN. First, we show that all other class II IN mutants have reduced affinities towards RNAs. Second, we show that WT IN tetramers but not dimers are able to bridge between separate target RNAs. Third, we show that all of the class II IN mutants tested exhibit substantially lower levels of RNA bridging activity. Taken together, these results strongly argue that IN multimerization is critical for the RNA binding/bridging ability of IN. Responses to the other suggestions/criticisms, which were also raised by other reviewers, are detailed below.

2) The crosslinking results in Figure 4A do not seem very convincing: there are high-MW IN species visible for all mutants. Perhaps some of them are hetero-multimer crosslinks (i.e. with other proteins in the virion?). SEC data, such as in panel B, are much clearer; I would suggest including more mutants in this analysis.Moreover, it is not clear how reproducible the cross-linking patterns are (and the WT in the third panel looks very different from the WTs in the first and second panels). The cross-linking results, to the extent that they can be interpreted, appear to be very different from the SEC profiles shown in Figure 4B. The authors conclude (subsection “IN multimerization plays a key role in RNA binding”) that the majority of class II IN mutants are predominantly monomeric in virions, yet the majority of the protein is found in high-MW aggregates. This issue needs to be addressed.

The reviewer raises important points and we addressed these concerns as follows. First, we have now included SEC profiles of other IN mutants (V165A, R199A, Q214L/Q216L and K258A), which show that many formed dimers but had defects in tetramerization in vitro (Figure 4B). Second, we have generated a table (Supplementary file 2), which summarizes published work on the multimerization properties of the class II IN mutants studied herein. Third, we have now included quantitation of IN multimers from three independent biological replicates for in virion crosslinking experiments in Figure 4A, which convincingly illustrate the multimerization defects of the IN mutants. Fourth, we have changed the text to clarify the experimental results.

We agree with the reviewer that in these experiments, a large fraction of the signal is found in high MW aggregates. While the multimerization experiments in virions are more physiologically relevant, they are also inherently more complex. For example, we cannot distinguish whether these high molecular weight species represent aberrant IN multimerization products, aggregates or crosslinking between IN and other viral proteins. Another possibility is that these high molecular weight species may not resolve well in this system on the 6% Tris-acetate gels, which unfortunately is a limitation of this approach. Notwithstanding, we believe that the quantitation of data from multiple experiments as provided now reveals consistent patterns in the multimerization properties of the class II mutants in virions.

Finally, we do not think that in vitro SEC profiles can be directly compared to the results obtained from in virion crosslinking experiments for several reasons. First, as chemical crosslinking is not 100% efficient and because the crosslinked proteins are separated under denaturing/reducing conditions, the in virion crosslinking assay may not report on the actual form and amount of IN multimers in virions. Second, as indicated above, crosslinking experiments done in virions are inherently more complex due to the presence of other viral proteins/RNAs that may be within close proximity of IN. Third, while the in vitro SEC experiments provide key information on the intrinsic properties of IN, they may not correlate 100% with the actual multimeric form of IN within the more complex setting of the virion. Thus, while each approach has limitations, we believe that they are complementary and highlight the importance of IN multimerization in RNA binding as a whole.

3) The key issue of whether the IN mutants are excluded or not from viral cores needs to be clarified. The sedimentation of the majority of IN mutants in the dense, CA-containing fractions in Figure 6A would seem to suggest that these mutant IN proteins are located within the viral capsid. However, in the cell-based uncoating assay, the IN mutants are degraded (Figure 7C) while a high percentage of capsids remain intact at the 2 hour time point (Figure 7B). These data suggest that the mutant IN proteins are excluded from the protective environment of the capsid and are thus rapidly degraded. Indeed, in the model figure (Figure 8), these mutants are depicted as being outside the cores. One possible solution to this conundrum is that the IN mutants are prone to aggregation under the conditions used in Figure 6 and thus co-sediment with CA without actually being inside the capsid. In any case, as currently presented, the data appear to be in contradiction and leave this important question unresolved.

The reviewer raises an important and interesting point that we also have thought of. We completely agree with the reviewer on the conclusions derived from the data and have clarified this point further in the text. Demonstrating precisely where within the virions class II IN mutants are located extends beyond the scope of this paper and future studies are warranted to study this process in detail. For example, we tried to directly assess where IN is located in virions by immunogold EM; however, these experiments suffered from low resolution and contrast due to the differences in this particular protocol compared to the standard EM procedures conducted in this study. This was compounded by the epitope inaccessibility for the antibodies used (we tried a total of 4 separate monoclonal and polyclonal antibodies) under the fixation conditions required for immunogold-EM experiments.

As an alternative, we took a biochemical approach to assess whether we can separate class II IN mutants from the CA lattice. To this end, we took advantage of previously defined capsid destabilizing mutation K203A. We reasoned that if the class II IN mutants are not bound to RNA and kept inside the CA lattice, CA destabilization would alter their migration pattern in biochemical fractionation experiments as done in Figure 7 of this manuscript. To our surprise, we found that CA destabilization did not alter the migration patterns of class II IN mutants. To completely rule out the possibility that class II IN mutants are still bound to RNA (which would explain their migration in dense fractions), we conducted similar experiments in viruses lacking NC, and thus are devoid of RNA, or in the presence of RNase throughout the fractionation protocol. Migration pattern of IN remained unaffected under these conditions either, making it difficult to assess whether class II IN mutants are located inside the CA lattice or are aggregated. These data are now included in the Figure 7—figure supplement 1. Based on these results, and as suggested by the reviewer, we favor the model that class II IN mutants are mislocalized outside of the CA lattice but likely form aggregates as they are prematurely degraded in target cells. This point is now clearly made in the text leading to this figure and in Discussion section.

4) The data presented in Figure 8 suggest that the class II IN mutations do not affect the assembly or stability of the CA lattice either in vitro or in target cells (but see Fontana et al., 2015). Yet, in the Figure 8 model, the capsid is shown as discontinuous or "broken" in the context of the IN mutants, a feature that would presumably significantly compromise capsid stability. The model thus appears to be inconsistent with the data.

We agree with the reviewer that this is an important point. We modified the model such that the CA lattice is complete as the data provided here do indeed suggest that CA assembly and stability are unaffected. The possible reasons for the discrepancy between our work and Fontana et al., 2015 are discussed in the text. The current text reads:

“Although this finding is in disagreement with the previously observed morphological aberrations of the CA lattice present in eccentric particles [26], it is possible that the biochemical experiments used herein lack the level of sensitivity required to quantitatively assess these aberrations or that the cryo-ET procedure impacts the structure of the CA lattice in particular in eccentric virions due to the absence of a packaged vRNP complex. A further possibility is that while the CA lattice in class II IN mutant virions may appear morphologically aberrant, they may still uncoat similar to WT virions in target cells.”

5) The difference between tetramers, dimers and monomers in Figure 4C is surprising. The species would be expected to re-equilibrate after SEC. Please comment.

We agree with the reviewer that this is an important point. While it is possible that the collected monomers and multimeric IN species can interconvert, we minimize this by subjecting the collected fractions from SEC immediately to the α screen assay. Thus, even if a fraction of dimers is converting to tetramers, it is unlikely to have a significant impact on the overall outcome of these experiments. The fact that we are observing a profound difference in the RNA binding properties of tetramers vs. dimers and monomers (for WT) indicates that the samples do not re-equilibrate at appreciable levels during the course of these experiments. Otherwise, we would observe similar levels of binding to viral RNAs.